# Tuning defects in oxides at room temperature by lithium reduction

Gang Ou[1,2], Yushuai Xu[1], Bo Wen[3], Rui Lin[2], Binghui Ge [4], Yan Tang[5], Yuwei Liang[1], Cheng Yang[1], Kai Huang[1], Di Zu[1], Rong Yu [6], Wenxing Chen [2], Jun Li[5], Hui Wu[1], Li-Min Liu[3,7] & Yadong Li[2]

Defects can greatly influence the properties of oxide materials; however, facile defect engineering of oxides at room temperature remains challenging. The generation of defects in oxides is difficult to control by conventional chemical reduction methods that usually require high temperatures and are time consuming. Here, we develop a facile room-temperature lithium reduction strategy to implant defects into a series of oxide nanoparticles including titanium dioxide ($TiO_2$), zinc oxide (ZnO), tin dioxide ($SnO_2$), and cerium dioxide ($CeO_2$). Our lithium reduction strategy shows advantages including all-room-temperature processing, controllability, time efficiency, versatility and scalability. As a potential application, the photocatalytic hydrogen evolution performance of defective $TiO_2$ is examined. The hydrogen evolution rate increases up to 41.8 mmol $g^{-1} h^{-1}$ under one solar light irradiation, which is ~3 times higher than that of the pristine nanoparticles. The strategy of tuning defect oxides used in this work may be beneficial for many other related applications.

[1] State Key Laboratory of New Ceramics and Fine Processing, School of Materials Science and Engineering Tsinghua University, 100084 Beijing, China. [2] Department of Chemistry and Collaborative Innovation Center for Nanomaterial Science and Engineering, Tsinghua University, 100084 Beijing, China. [3] Beijing Computational Science Research Center, 100193 Beijing China. [4] Beijing National Laboratory for Condensed Matter Physics, Institute of Physics, Chinese Academy of Science, 100190 Beijing, China. [5] Department of Chemistry and Key Laboratory of Organic Optoelectronics and Molecular Engineering of Ministry of Education, Tsinghua University, 100084 Beijing, China. [6] National Center for Electron Microscopy in Beijing, School of Materials Science and Engineering, Tsinghua University, 100084 Beijing, China. [7] School of Physics, Beihang University, 100191 Beijing, China. These authors contributed equally: Gang Ou, Yushuai Xu, Bo Wen. Correspondence and requests for materials should be addressed to H.W. (email: huiwu@tsinghua.edu.cn) or to L.-M.L. (email: liminliu@buaa.edu.cn) or to Y.L. (email: ydli@tsinghua.edu.cn)

D efect engineering of oxide materials has been a major focus in materials science, since the fundamental physical and chemical properties of oxide materials greatly rely on their defect structures[1–7]. For example, the optical absorption of oxides can be greatly enhanced by implanting defects[8,9]. Besides, defects have also been reported to enhance catalytic activity by taking advantage of defect sites as active centers[10–12]. Therefore, the science and technology behind the tuning of defect structures in oxide materials have attracted increasing attention in the past several decades[13–17]. Recently, Li et al. reported a chemical leaching method to prepare jagged nanowires with defective sites at the surface. In their work, they found highly enhanced oxygen reduction reaction performance[12]. Chen et al. reported defective "black" TiO₂ with significantly enhanced photocatalytic activity by hydrogenating pristine $TiO_2$ nanoparticles in a 20 bar $H_2$ atmosphere at 200 °C for 5 days[18]. Reductive metals have also been introduced to tune the defects in $TiO_2$ at high temperatures, such as Mg and Al[19,20]. In addition, Nakajima et al. reported defective $TiO_2$ by high energy laser irradiation, providing another strategy to tune the defects in oxide materials[21,22]. Nevertheless, controllable defect engineering in oxide materials under ambient conditions remains challenging. Meanwhile, defect engineering has emerged as an effective strategy to tune the electronic structure of metal oxides, which is vital for many applications[2,5,23–26].

According to Ellingham diagrams[27], lithium metal, which possesses high reductive activity, can potentially reduce a significant number of metal oxides at room temperature. Therefore, Li reduction may provide exciting opportunities to partially reduce a series of oxide materials at room temperature, removing oxygen, generating oxygen vacancies and related defects in these materials.

Here, we show that by simply grinding pristine oxide powders with lithium metal powder, followed by an acid leaching process for the removal of lithium oxides from the mixture, a series of oxide materials including $TiO_2$, ZnO, $SnO_2$, and $CeO_2$ are implanted with a high concentration of defects. Moreover, we demonstrate that the $TiO_2$ nanoparticles with tunable defects demonstrate high activity and long stability as a photocatalyst for the hydrogen evolution reaction.

## Results

**Thermodynamic calculations.** To prove that lithium can effectively reduce $TiO_2$, $SnO_2$, ZnO etc., as represented in Eq. (1), the reaction enthalpies were calculated. As shown in Supplementary Table 1, the enthalpy changes of reaction represented by Eq. (1) are all negative, which means the reactions are exothermic. Furthermore, for example, $\Delta G$ for the reaction $4Li + TiO_2 \rightarrow Ti + 2Li_2O$ is $-233.6$ kJ mol$^{-1}$ at 298 K, indicating the reaction can occur spontaneously at room temperature.

$$MO_y + 2x Li \rightarrow MO_{y-x} + x Li_2O. \qquad (1)$$

**Synthesis and characterization of defective oxides.** We applied the lithium reduction method to create defects in $TiO_2$ nanoparticles (P25, Degussa, commercially available) since $TiO_2$ has been widely studied as a high performance photocatalyst in recent decades[28,29]. $TiO_2$ and lithium powders were weighed and placed in a mortar, followed by the addition of dimethyl carbonate (DMC) as a dispersant. The mixtures were ground up and then removed by dissolving the generated lithium oxide with dilute hydrochloric acid (HCl). The detailed process is shown in Supplementary Fig. 1. After centrifuging and washing, the obtained powders were dried for characterization. As shown in Fig. 1a, the lithium-reduced $TiO_2$ samples appear in different colors ranging

from blue to black, in marked contrast to the white pristine $TiO_2$ powders. The color change of $TiO_2$ may be ascribed to the enhanced absorption of visible light and thus we named them "Black" $TiO_2$. Figure 1b shows the identical crystalline structure with the anatase phase in majority and that of rutile in minority for various $TiO_2$ samples, which demonstrates that our lithium reduction method does not alter the intrinsic crystal structure of the $TiO_2$ nanoparticles. As shown in the Raman spectra in Supplementary Fig. 2, the Raman vibration band of pristine $TiO_2$ powders exhibits an identical configuration to that of the lithium-reduced $TiO_2$ except for the broadening of the $E_g$ peak at ~148 cm$^{-1}$ (Supplementary Fig. 3). It is found that the particle size of lithium-reduced $TiO_2$ is almost the same as that of pristine $TiO_2$, based on the morphology characterized by scanning electron microscopy (SEM, Supplementary Fig. 4), which indicates that the room-temperature conditions do not cause further growth of the grains. To demonstrate that our approach can be used as a versatile method to generate defects in other oxides, we prepared similar "Black" ZnO, $SnO_2$, and $CeO_2$ nanoparticles by the lithium reduction route (Fig. 1c) without changing their crystal structures (Fig. 1d–f) or particle size (Supplementary Fig. 5).

To evaluate the defects of lithium-reduced $TiO_2$, X-ray photoelectron spectroscopy (XPS) of elemental O and Ti are provided. Three typical XPS peaks at ~530.6, 531.9, and 532.8 eV (labeled as O1, O2, and O3) in Fig. 2a can be attributed to lattice oxygen, oxygen defects, and surface-adsorbed oxygen species, respectively[18,30]. As observed in Fig. 2a, the content of oxygen defects (O2) in lithium-reduced $TiO_2$ exhibits a significant increase compared with that of pristine $TiO_2$, suggesting that the active lithium can efficiently remove partial oxygen elements and then generate more oxygen vacancies in the lattice of $TiO_2$ powders. In addition, the content of oxygen defects in lithium-reduced $TiO_2$ powders increased noticeably with increasing lithium metal powders, which indicates the facile use of lithium to tune the defect content in $TiO_2$ powder. Furthermore, the Ti 2p spectra of $TiO_2$ powder also demonstrates that infrequent $Ti^{3+}$ was produced from the generation of oxygen vacancies[30] (Fig. 2b and Supplementary Fig. 6). Since XPS analysis is rather insensitive to the detection of $Ti^{3+}$, we applied electron paramagnetic resonance (EPR) measurements to study the titanium defects in $TiO_2$ powders. The indicator values of g-peaks at 1.973 and 2.002 correspond to $Ti^{3+}$ and oxygen vacancies in the lattice, respectively[30,31] (Fig. 2c). Consistent with the results of the XPS spectra, the content of $Ti^{3+}$ and oxygen vacancies in lithium-reduced $TiO_2$ are significantly higher than those in the pristine powder. It is important to note that no elemental lithium was found in either the pristine $TiO_2$ or the lithium-reduced $TiO_2$ powders as determined by Li 1s spectra[32] (Supplementary Fig. 7), which suggests that lithium oxides can be thoroughly removed by acid treatment. Further characterization by transmission electron microscopy (TEM) in Fig. 2d and e reveals that the pristine $TiO_2$ has no disordered domains, but after treatment with 5% Li a disordered layer with a thickness of 2 nm was observed. We attribute this to the deprivation of surface lattice oxygen by elemental lithium during the contact reaction process. Interestingly, we also found that the specific surface area of $TiO_2$ nanoparticles increased from 49.88 to 55.28 m$^2$ g$^{-1}$ after lithium reduction treatment, which can be attributed to the generated defective surface structure.

Moreover, we investigated the conductivity of the $TiO_2$ nanoparticles (Supplementary Fig. 8), obtaining significantly enhanced conductivity after lithium reduction treatment due to the generated disordered surface layer. It is well-known that the existence of the $Ti^{3+}$ can enhance the conductivity of $TiO_2$[33–35]. As shown in Supplementary Fig. 9, the existence of oxygen vacancies produces excess electrons, which transforms $Ti^{4+}$ into

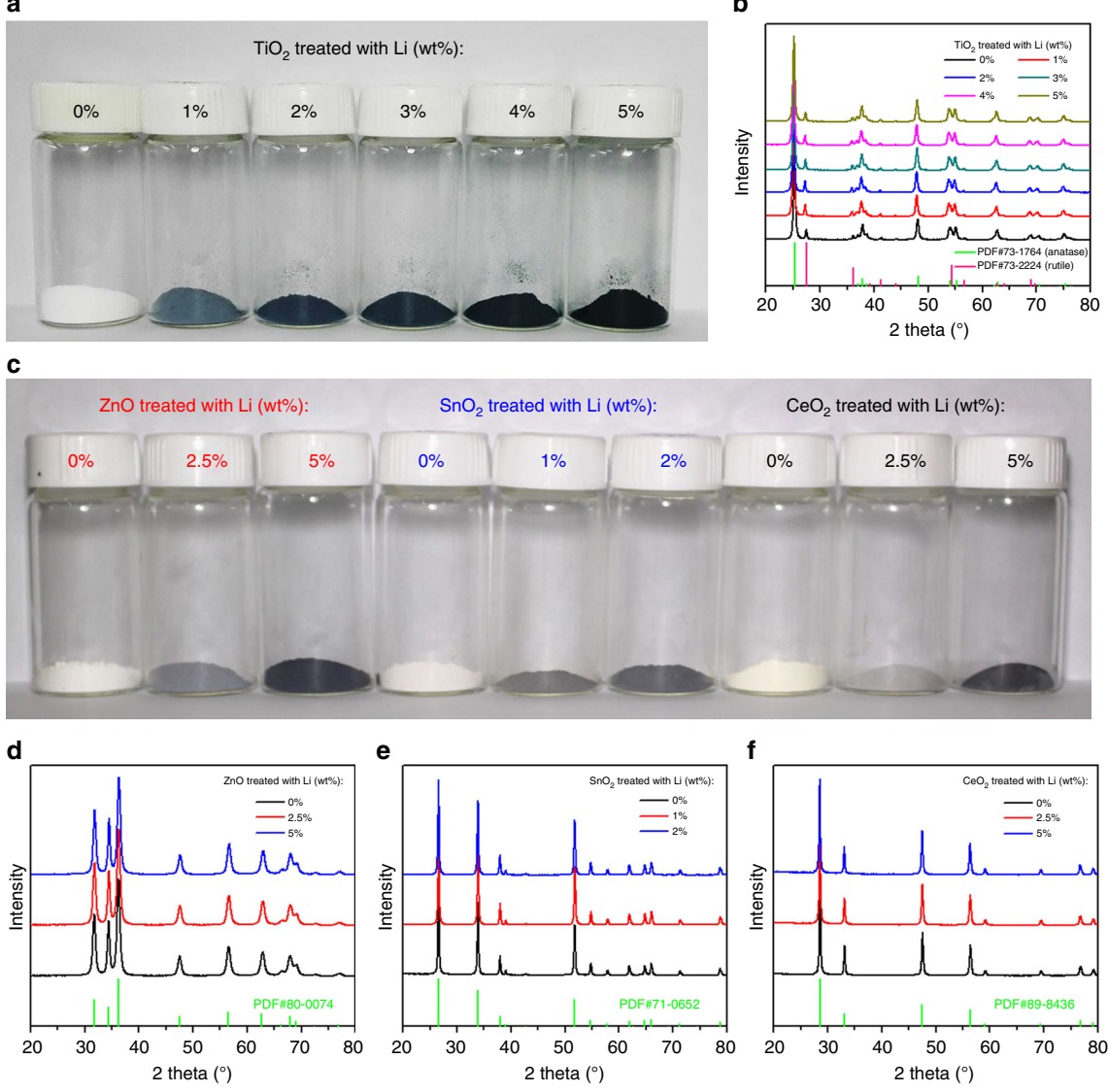

**Fig. 1** Pristine and lithium-reduced oxide nanoparticles. **a** Photograph of $TiO_2$. **b** XRD patterns of $TiO_2$. **c** Photograph of ZnO, $SnO_2$, and $CeO_2$, respectively. **d**–**f** XRD patterns of ZnO, $SnO_2$, and $CeO_2$, respectively

$Ti^{3+}$. Meanwhile, the concentration of $Ti^{3+}$ greatly depends on the amount of oxygen vacancies ($O_v$). The excess electrons can freely hop at room temperature[36–38], which is likely the main reason for the high conductivity of the reduced $TiO_2$. Beyond $TiO_2$, the "Black" ZnO, $SnO_2$, and $CeO_2$ exhibit similar defects structure compared with $TiO_2$. We observed that the content of oxygen defects in all the oxides significantly increased after lithium reduction treatment (Supplementary Fig. 10), indicating that the oxygen defects have been successfully implanted in the oxides. At the same time, the binding energy of the metal cations in the oxides slightly decreased after lithium reduction treatment due to the implanted defects (Supplementary Fig. 11). In addition, no peaks related to Li were detected in the lithium-reduced oxides[32] (Supplementary Fig. 12), which further confirms that all the remaining lithium oxides were fully removed during the acid leaching process.

In order to investigate the effect of Li metal powder particle size on the reduction of the oxides, we prepared defective $TiO_2$ nano-powders reduced by 5 wt% Li powder and a single piece of Li. Here, the average diameter of the Li powder particles was ~30 μm (Supplementary Fig. 13). As shown in Supplementary Fig. 14,

it is apparent that the defective $TiO_2$ powders produced with the Li powder and the Li piece exhibit the same color depth, indicating their defective content to be consistent. In addition, we also compared different content of DMC on the reduction of $TiO_2$ nano-powder. As shown in Supplementary Fig. 15, it can be seen that the color of the defective $TiO_2$ did not change with the dispersant content from 1:10, 1:20 to 1:30. Our first-principles calculations show that the formation energy of an oxygen vacancy ($O_v$) in $TiO_2$ is 3.77 eV per oxygen, while the corresponding formation energy of losing one oxygen atom in DMC is 4.80 eV per oxygen, which is 1.03 eV larger than the corresponding formation energy through capturing oxygen from $TiO_2$. Thus, Li prefers to capture oxygen from $TiO_2$ instead of DMC.

From the above results, we are confident that uniform defects were successfully implanted at the surface of the oxide nano-powders by our lithium reduction strategy, which can be ascribed to several reasons. Firstly, the lithium metal can react with oxides at room temperature according to the Ellingham diagrams and our calculated results. Thus the lithium metal can draw O from the oxides to generate $Li_2O$ as well as defective oxides during the grinding process. Secondly, the lithium can be continuously

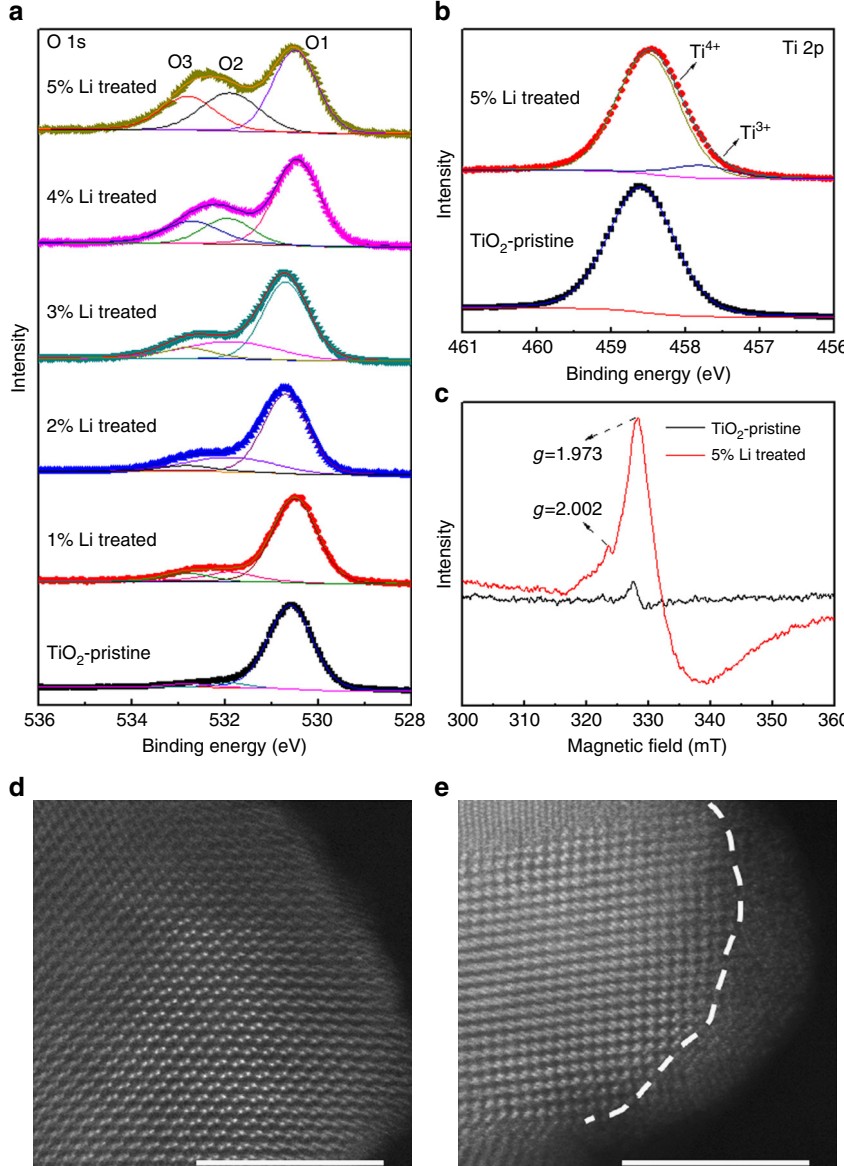

**Fig. 2** Defect characterization of pristine and lithium-reduced $TiO_2$ nanoparticles. **a**, **b** XPS spectra. **c** EPR spectra. **d**, **e** High angle annular dark field (HAADF) images of pristine and 5% Li-treated $TiO_2$. Scale bars, 5 nm

extended during the grinding process because of its high ductility, resulting in the constant exposure of fresh lithium to react with oxides, until the lithium metal fully reacts. Lastly, the lithium powders and oxide nano-powders can be uniformly mixed with the addition of DMC, which is beneficial for the uniform implantation of defects at the surface of the oxide nano-powders during the grinding process. In addition, the reaction between Li and $TiO_2$ occurs at the interface, which captures the oxygen atoms at the $TiO_2$ surface. Although the oxygen vacancies are initially generated at the surface, they prefer to diffuse into the subsurface or the inner layer (Supplementary Fig. 16). The subsurface $O_v$ is more stable by ~0.13 eV than the one at the surface, as suggested in previous experiments and theory[39–41]. The calculated diffusion energy barrier is ~0.7 eV (Supplementary Fig. 16), which indicates that the $O_v$ has the tendency to diffuse from the surface to the inner layer region at room temperature.

As discussed above, the defective oxides often exhibit superior properties than their pristine counterparts for a range of applications[2,5,23–26]. Therefore we expect the defective metal

oxides fabricated in this work to have many potential applications. In the following, as examples of potential applications, we examined the photocatalytic degradation of organic pollutants and the hydrogen evolution of our defective $TiO_2$ nano-powder.

**Photocatalytic properties of defective $TiO_2$ nano-powder.** Supplementary Fig. 17 shows the UV-Vis-NIR absorption spectra of $TiO_2$ powders at different reduction levels to explain the optical absorption change induced by the formation of defects. It is found that the absorption of visible and near-infrared light for lithium-reduced $TiO_2$ was enhanced significantly compared to that of pristine $TiO_2$ powder, and there is a positive correlation between the enhancement and the lithium metal content. According to the fitted optical bandgap, as given by the Tauc equation[42], the bandgap of the $TiO_2$ narrowed from 3.3 to 3.1 eV after treatment with lithium (Supplementary Fig. 18), suggesting that the introduction of defects in reduced $TiO_2$ can slightly narrow the bandgap and also widen the range of light absorption.

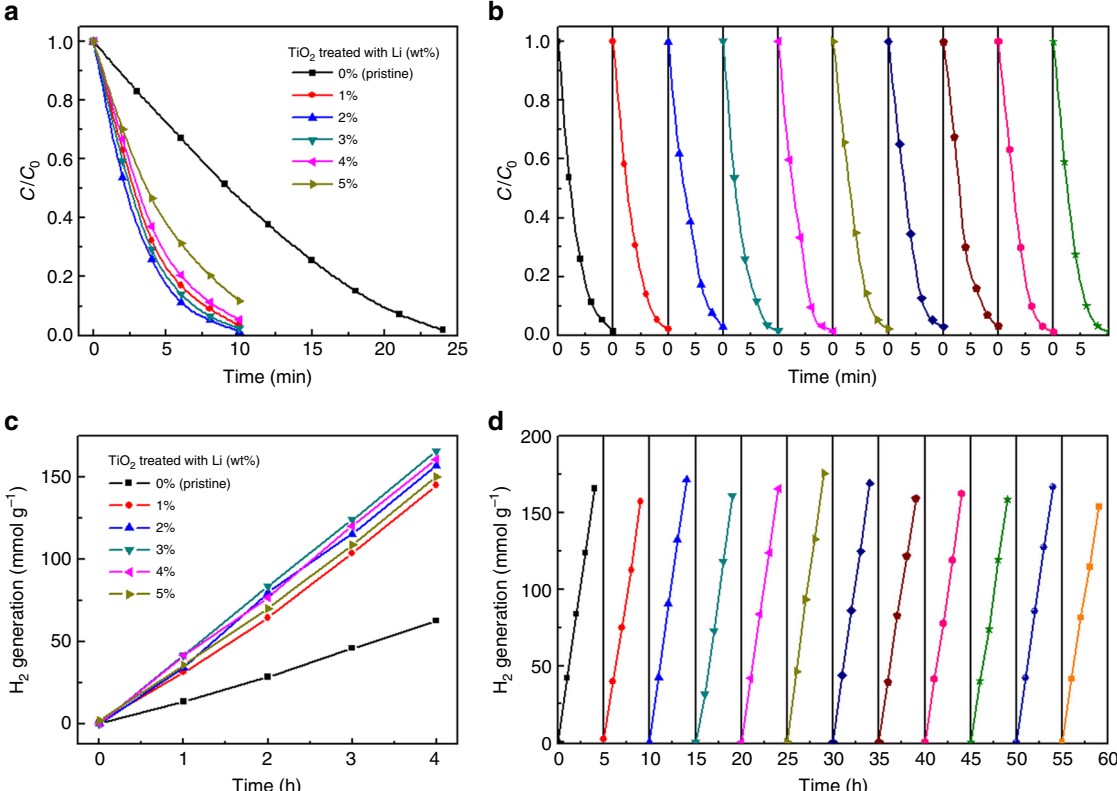

**Fig. 3** Photocatalytic properties of pristine and lithium-reduced TiO$_2$ nanoparticles. **a**, **b** Photocatalytic activity and stability for degradation of RhB. **c**, **d** Photocatalytic activity and stability for hydrogen evolution

In addition to optical absorption properties, the separation and migration of non-equilibrium state carriers are also affected by the defect formation. The fluorescence in solid steady state could shed light on the transportation and recombination kinetics of photogenerated electron−hole pairs qualitatively. Supplementary Fig. 19 displays that both the pristine and defective TiO$_2$ show their emission peaks at 412 nm when excited by the 330 nm light. The lower fluorescence emission intensity of defective TiO$_2$ implies that its recombination of excited state electron−hole pairs is less severe than that of pristine TiO$_2$, which may endow defective TiO$_2$ with a longer lifetime of carriers. Transient state fluorescence tests confirm our prediction successfully (Supplementary Fig. 20). The fluorescence decay profile showed that the lifetime of photogenerated carriers in defective TiO$_2$ (280 ps) is longer than that of pristine TiO$_2$ (230 ps), which proves that the defects promote the separation of photogenerated carriers. Furthermore, the optical absorption of ZnO, SnO$_2$, and CeO$_2$ nanoparticles was also measured, showing greatly enhanced absorption (Supplementary Fig. 21) and a marginally narrowed bandgap (Supplementary Fig. 22) after the lithium reduction treatment-induced defects implantation in these oxides.

Theoretically, the defective TiO$_2$ photocatalyst with wider solar spectrum response range and longer charge carrier lifetime will exhibit higher activity. In line with our deduction, we observed a significant improvement of rhodamine B (RhB) photocatalytic degradation efficiency in lithium-reduced TiO$_2$ compared with that in pristine TiO$_2$ (Fig. 3a). The time taken for TiO$_2$ photocatalyst to decompose RhB under solar light decreased from 24 to 10 min after the lithium reduction treatment. Interestingly, the photocatalytic degradation activity demonstrated first an increase and then a decrease with increasing lithium content, and attained optimization after treatment with 2% Li. Therefore, we repeated the degradation of RhB nine times

based on the 2% Li-reduced TiO$_2$ nanoparticles (Fig. 3b). We clearly observed that the lithium-reduced TiO$_2$ nanoparticles remain constant after nine cycles' degradation of RhB, suggesting their excellent stability. Afterwards, we tested the photocatalytic hydrogen evolution activity. Compared with pristine TiO$_2$, the lithium-reduced TiO$_2$ revealed significantly enhanced photocatalytic activity (Fig. 3c). The photocatalytic hydrogen evolution rate increased from 13.4 mmol g$^{-1}$ h$^{-1}$ (pristine TiO$_2$) to 41.8 mmol g$^{-1}$ h$^{-1}$ (3% Li-reduced TiO$_2$) under a full solar light (containing ultraviolet, visible and near-infrared light at an irradiation density of 100 mW cm$^{-2}$). In addition, the photocatalytic hydrogen evolution activity also demonstrated an initial increase and then decrease with increasing lithium content, and achieved its peak after treatment with ~3% Li, which is consistent with the above photocatalytic decomposition activity. We noticed that the defects have various impacts on the photocatalytic property; that is, the photocatalytic property is proportional to the content of defects up to a certain point, due to the enhanced light absorption and active sites. However, excess defects are detrimental to the photocatalytic activity since they can also act as charge recombination centers[31,43]. Subsequently, we repeated the hydrogen evolution experiments 11 times based on the 3% Li-treated TiO$_2$ nanoparticles (Fig. 3d). It is apparent that the lithium-reduced TiO$_2$ demonstrates consistently excellent activity after long-term cycles, which also indicates the superior stability of the defective TiO$_2$ and the implanted defects within them. Based on the above photocatalytic results, we suggest that the enhanced photocatalytic properties can be attributed to the implanted defects in the TiO$_2$ nanoparticles. In order to prove this, we further characterized the photocatalytic degradation activity of lithium-reduced TiO$_2$ nanoparticles after annealing at 400 °C for 1 h to remove the generated defects (Supplementary Fig. 23). The results showed that the photocatalytic performance

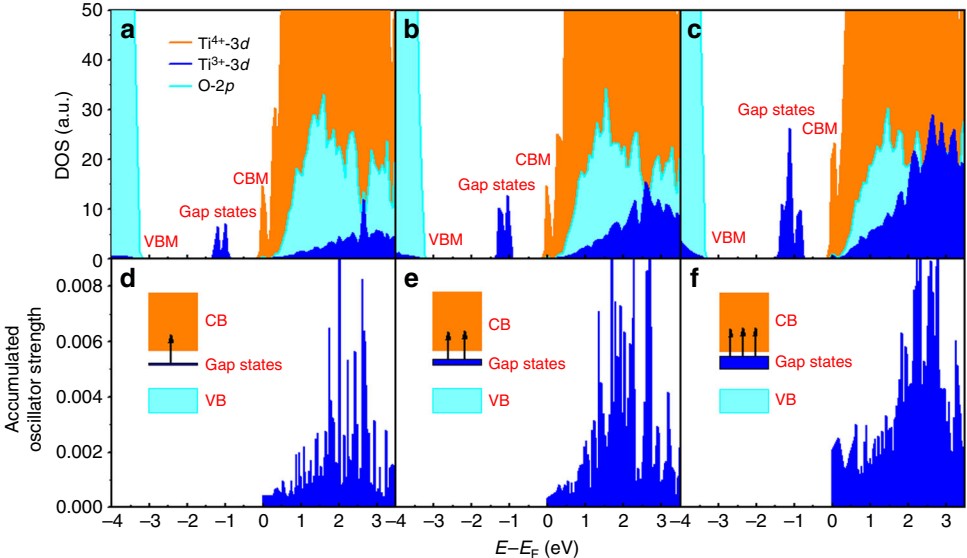

**Fig. 4** Density of states (DOS) and corresponding accumulated oscillator strength of defective anatase $TiO_2$. **a–c** DOS of $TiO_2$ surface with $1O_v$, $2O_v$, and $4O_v$ per slab, respectively. **d–f** Oscillator strength of $TiO_2$ surface with $1O_v$, $2O_v$, and $4O_v$ per slab, respectively. The oscillator strength was calculated for transitions from gap states to conduction band (CB). The orange, blue, and light green area in the DOS stand for projected DOS of $Ti^{4+}$ ions, $Ti^{3+}$ ions, and $O^{2-}$ ions, respectively. The gap state, valence band maximum (VBM), and conduction band minimum (CBM) are indicated in the DOS. Accumulated oscillator strength is calculated by summing the component data of $y$ and $z$ direction in Supplementary Fig. 24

of defective $TiO_2$ significantly declined after annealing. Furthermore, we also compared the photocatalytic hydrogen evolution activity of the lithium-reduced $TiO_2$ nanoparticles with previously reported works (Supplementary Table 2), which indicates that our defective $TiO_2$ by lithium reduction possesses superior photocatalytic activity.

**Theoretical calculations**. To understand the origin of the change in electronic and optical properties of lithium-reduced $TiO_2$ nanoparticles, the electronic structure was predicted by first-principles calculations. Firstly, the density of states (DOS) for reduced $TiO_2$ with different $O_v$ concentrations was examined (Fig. 4a–c). As indicated in Fig. 4, the valence band maximum (VBM) was mainly composed of O-2$p$ orbitals, and the conduction band minimum (CBM) mainly consisted of Ti-3$d$ orbitals. Interestingly, the change of $O_v$ concentrations ($1O_v$, $2O_v$, and $4O_v$) does not affect the positions of either VBM or CBM, so the intrinsic bandgap is not affected by $O_v$. As found above, the intrinsic band gap of $TiO_2$ is only changed from 3.3 to 3.1 eV after inducing $O_v$, which is consistent with the theoretical results.

Considering that $O_v$ does not affect the intrinsic bandgap, it is natural to argue why the light absorption of $TiO_2$ containing $O_v$ is greatly enhanced compared with that of pristine $TiO_2$. It has been previously reported that excess electrons can be introduced by $O_v$[44]. Meanwhile, excess electrons can be easily trapped by the $Ti^{4+}$ ions to form $Ti^{3+}$ ions, and the gap states induced by $Ti^{3+}$ ions exist within the intrinsic band gap[45]. The projected DOS of $Ti^{4+}$, $Ti^{3+}$ and $O^{2-}$ ions are shown in Fig. 4a–c, respectively. The orange, blue, and light green areas represent the corresponding projected DOS of $Ti^{4+}$ ions, $Ti^{3+}$ ions, and $O^{2-}$ ions, respectively. As shown in Fig. 4a–c, the peak of gap states (the blue peaks) gradually become larger and broadened with the increase of $O_v$ concentration. Meanwhile, the intensity of PDOS of $Ti^{3+}$ ions in the CB region, especially the peak ~2 eV above CBM was also greatly enhanced[46]. In fact, the CBM of $TiO_2$ is composed of empty $t_{2g}$ orbitals, so when the excess electrons exist in the $TiO_2$ system, each of them will firstly fill one of the three nearly degenerated $t_{2g}$ orbitals and leaves two unoccupied states.

The filled $t_{2g}$ orbitals become the gap states, while the other two unoccupied orbitals of $t_{2g}$ move to higher energy levels, corresponding to the resonance of the gap states, consistent with the Jahn−Teller effect[46].

In order to know whether the gap states affect photoabsorption, the oscillator strength of transitions from gap states to the CB were further calculated for $Ti^{3+}$. As shown in Fig. 4d–f, some peaks at ~2 eV appear for the oscillator strength of the transition from gap states to the CB. Such results clearly suggest that the gap states induced by $O_v$ can enhance the photoadsorption through $d$−$d$ transitions between $Ti^{3+}$ ions and its resonance[47]. It should be noted that the oscillator strength of $Ti^{4+}$ does not have such kind of feature. Most importantly, with the increase of $O_v$, the peak of the resonance states around 2 eV gradually becomes much more apparent (Fig. 4e, f). Since the gap states have a direct relationship with the $O_v$ defects, the gap states can be greatly broadened and even extended to the CBM[48]. As a result, the energy difference between gap states and resonance states in CB dramatically decreases with the increase of $O_v$, which falls into the energy range of the visible light region. This is why reduced $TiO_2$ could absorb visible light and its apparent color changes from blue to dark with the increase of $O_v$ concentration.

**Discussion**

From the above results, it can be clearly seen that the lithium-reduced $TiO_2$ nanoparticles show significantly enhanced photocatalytic properties compared with pristine $TiO_2$, which may be correlated with the greatly enhanced light absorption, improved conductivity, surface disorder layer, implanted oxygen vacancies, and generated $Ti^{3+}$. In particular, the improved light absorption and photocatalytic properties are attributed to the $Ti^{3+}$ of the defective $TiO_2$. Furthermore, metallic conduction can be achieved at the crystalline−amorphous interface of the defective $TiO_2$ nanoparticles, which enhances the electronic transport properties of $TiO_2$[34]. Another important factor is the implanted oxygen vacancies and/or surface disorder, which also plays a key role in increasing the photocatalytic activity. The donor density of $TiO_2$ is enhanced by introducing oxygen vacancies since they can act as

electron donors[49], which improves charge transport and shifts the Fermi level towards the conduction band[50], facilitating charge separation and improving the incident photon-to-current efficiency (IPCE) in the UV region[51]. Finally, the generated $Ti^{3+}$ in the defective $TiO_2$ can reduce the recombination of the photogenerated electron–hole pairs and thus improve the photocatalytic activity of the lithium-reduced $TiO_2$ nanoparticles[19].

We note that the contact reaction between the Li and $TiO_2$ powders proceeds very quickly in principle, since Li can diffuse quickly in the oxides. However, we observed from Supplementary Fig. 1 that the color of the mixed powders turned darker after grinding for longer times, indicating that the defects formation requires long reaction times. This is because lithium oxide passivation layers (like $Li_2O$), formed at the interface between Li and defective oxide powders, buffer the reaction between Li and the oxides. Therefore, only a small amount of Li diffuses into the oxides at the Li/oxide interface. However, with continuous hand-grinding, the oxide passivation layer was destroyed by the shearing and friction forces between particles, with fresh lithium exposed to react with fresh oxide surfaces. After lengthy grinding process, the lithium oxide passivation layer was cyclically formed and destroyed, and the lithium finally reacted entirely with oxide powders, resulting in the formation of uniform defective oxide powders. While we used hand-grinding here to demonstrate the successful Li-reduction and defect-implanting chemistry, ball-milling or machine grinding could also be introduced to speed up the reactions, making the process more controllable and time-efficient in the future study. Thus, we suggest that almost all of our lithium-reduced oxide nano-powders have a defective structure based on the following reasons: (1) Besides TEM, the defect structures have also been confirmed by XPS spectra and EPR spectra (Fig. 2 and Supplementary Fig. 10), which provided the average overall analysis of the materials. (2) The defective oxide powders have demonstrated significantly enhanced photocatalytic properties after lithium reduction. In addition, the materials undergo uniform color changes from white to black. (3) Our experimental procedure ensures that all substrates are evenly involved in the reaction. Firstly, we chose oxide nanoparticles and Li powders instead of micro oxide powders and Li foil as the substrates to avoid possible uneven contact. Secondly, we added solvent (DMC) to enhance their contact while grinding. Lastly, we applied long-time (~1 h) grinding process to allow the oxide particles to evenly attach and react with the lithium powder. (4) Our experimental and theoretical discussion also pointed out that the Li atoms have a high reaction rate with oxides; that is to say, the defect implanting reaction is fast, which further assists the formation of uniform defects in the nanoparticles.

## Methods

**Materials preparation.** The used raw materials are commercially available; they are $TiO_2$ (99.9%, Degussa), ZnO (99.9%, Aladdin), $SnO_2$ (99.9%, Aladdin), $CeO_2$ (99.5%, Aladdin), and Li powders (99.9%, Cellithium, China). A certain amount of oxide material was weighed and put in a mortar, then lithium powders and DMC (99%, Aladdin) were added acting as reducing agent and dispersant, respectively. In order to protect the lithium powders from oxidizing by air, the milling process was conducted in a glove box filled with Ar. The materials were carefully mixed by hand, grinding with a speed of ~2 laps per second for 1 h and removed to dissolve the generated lithium oxide by dilute HCl. After centrifuging and washing by deionized water three times, the obtained oxide powders were dried for characterization. In this study, different contents of lithium metal powder were added to the oxide nanoparticles to engineer the defect content. It is noted that although highly active Li metal powders were used in the experiment, there is minimal danger in the whole experimental process during regular operation. Firstly, the lithium powders used in this work are mass-produced in industry and treated by surface passivation. Furthermore, the lithium powders were put into a sealed glove box full of argon for safe storage and usage. Lastly, only a small amount of lithium powder (less than 5 wt%) is required by our method.

**Structural characterization.** The crystal structures were investigated by X-ray diffraction (XRD, D/max-2500, Rigaku). The morphology and microstructure were analyzed by SEM (MERLIN VP Compact, Zeiss) and aberration-corrected high-resolution transmission electron microscopy (JEM-ARM200F, JEOL). The X-ray photoelectron spectra of the $TiO_2$ samples were measured at beamline BL10B of Hefei National Synchrotron Radiation Laboratory (NSRL) in China. The binding energies of ZnO, $SnO_2$, and $CeO_2$ were characterized using an X-ray photoelectron spectrometer (XPS, Escalab 250Xi, Thermo Fisher Scientific). The absorption spectra were collected by UV-Vis-NIR spectrophotometer (UV-2600, Shimadzu). The defects were detected by EPR spectroscopy (FA-200, JEOL). The Raman spectra were measured on a microscopic confocal Raman spectrometer (Raman, LabRAM HR800, HORIBA Jobin Yvon) using a 532 nm laser as the excitation source. The specific surface area was measured by a surface area analyzer (QuadraSorb SI, Quantachrome). The solid steady-state fluorescence and transient state fluorescence spectra were tested by fluorescence spectrometer (FLS920, Edinburgh Instruments). The impedance spectra were measured by an electrochemical workstation (PGSTAT204, Autolab) with a three-electrode system in 0.5 M $H_2SO_4$ solution, in which Ag/AgCl electrode and carbon rod were used as reference electrode and counter electrode. The $TiO_2$ nanoparticles were dispersed ultrasonically in deionized water and ethanol (volume ratio = 1:1) mixed solution with Nafion 117 (Aldrich) as binder for 30 min and then dipped on carbon fiber paper (Toray) with a loading of 0.1 mg cm$^{-2}$ as working electrode.

**Photocatalytic measurements.** The photocatalytic activity was performed under simulated solar light (PLS-SXE 300C, Perfectlight) with AM1.5G at a light density of 100 mW cm$^{-2}$. The solution with RhB ($2.5 \times 10^{-5}$ mol L$^{-1}$) and $TiO_2$ nano-particles (1 mg ml$^{-1}$) was ultrasonicated for 30 min, magnetically stirred for 1 h in the dark and then irradiated under a full solar light. The concentration of RhB was measured by a spectrophotometer (UV-2600, Shimadzu). In the photocatalytic degradation cycle experiments, the $TiO_2$ nanoparticles were centrifuged, washed and dried for the next cycle, and the process was repeated nine times. In the photocatalytic hydrogen evolution experiments, Pt nanoparticles were applied as a co-catalyst by adding chloroplatinic acid in the solution with $TiO_2$ nanoparticles under UV light for 1 h. Methanol (10 ml), 40 ml deionized water, and 1.25 mg $TiO_2$ nanoparticles (1% Pt) were added in a closed glass beaker. The solution was irradiated by a full solar light under Ar atmosphere and the generated $H_2$ was determined by a gas chromatographer (GC-7920, CEAULIGHT). For the stability experiments, the same solution was continually irradiated for 11 cycles. Before each run, extra methanol was added to make the volume of the solution reach to 50 ml.

**Calculation details.** First-principles calculations were performed with the CP2K/Quickstep package, in which a hybrid Gaussian and plane-waves was applied[52]. In such method, the valence electrons were expanded in terms of Gaussian functions with molecularly optimized double-$\zeta$ polarized basis sets (m-DZVP)[53]. For the auxiliary basis set of plane waves, a 320 Ry cut-off was used. The generalized gradient approximation of Perdew, Burke, and Ernzerhof exchange-correlation functional was also chosen for calculations[54]. Hubbard U (3.5 eV) correction was used to calculate the strong exchange interaction of Ti 3$d$ orbital as well as O 2$p$ orbital[55–57]. Core electrons were described with norm-conserving Goedecker, Teter, and Hutter (GTH) pseudopotentials[58]. All atoms in the slab were relaxed until the maximum residual force is less than 0.02 eV Å$^{-1}$. To estimate the difficulty of $O_v$ diffusion from surface to subsurface, the transition states along the reaction pathways are searched by the Climbing Image Nudged Elastic Band (CI-NEB) approach[59].

In order to know whether lithium can oxidize $TiO_2$, $SnO_2$, ZnO etc. as represented in Eq. (2), the reaction enthalpy was calculated. Enthalpy $H$ equals the internal energy $U$ plus pressure-volume work pV, that is, $H = U + pV$. Because all the energies are calculated in the solid state, the pressure-volume effect can be neglected. The reaction enthalpy is contributed mainly by the internal energy of the reactants or products. Here, the enthalpy change $\Delta H$ is calculated using the following formula:

$$\Delta H(\text{Li}) = E_{\text{MO}-\text{O}_v} + 2 * E_{\text{Li}} - E_{\text{MO}} - E_{\text{Li2O}}, \quad (2)$$

where $E_{\text{MO}-\text{O}_v}$, $E_{\text{Li}}$, $E_{\text{MO}}$, and $E_{\text{Li2O}}$, are the total energy of bulk metal-oxide with $O_v$, total energy of a lithium atom in bulk phase, total energy of perfect bulk metal-oxide and total energy of a lithium oxide unit in bulk phase, respectively. The formation energy $E_{\text{form}}$ is calculated by $E_{\text{form}} = E_D - E_P + 0.5 * E_{O2}$, where $E_D$, $E_P$ and $E_{O2}$ are the total energy of defect slab, perfect slab and $O_2$, respectively.

The anatase-$TiO_2$ (110) surface was modeled using a repeated slab geometry with four $TiO_2$ tri-layers and a (1 × 4) (10.35 Å × 15.21 Å) surface supercell. The vacuum separation between slabs was around 15 Å. $O_v$ defects with different concentrations in the reduced anatase were modeled by removing one, two or four oxygen atoms from the subsurface of the supercell used in this work; see Supplementary Fig. 16. Due to the large computational resources for CI-NEB, a three tri-layers slab was adopted for the estimation of $O_v$ transition.

The oscillator strength was calculated using the following equation[60]:

$$f_{cv}^{\mu} = \frac{2}{m_e(E_c - E_v)} \left| \langle v | p_\mu | c \rangle \right|^2 \quad (3)$$

where $f_{cv}^{\mu}$ is the oscillator strength in the $\vec{e}_{\mu}$ polarized direction. $\langle v|$ denotes Kohn−Sham orbitals corresponding to VBM state or gap state and $|c\rangle$ denotes Kohn−Sham orbitals corresponding to unoccupied MOs above $E_F$. $E_c$ and $E_v$ correspond to the eigenvalue of $|c\rangle$ and $\langle v|$'s orbital, respectively. $p_{\mu}$ is the momentum along $\vec{e}_{\mu}$ direction.

**Data availability**. The datasets generated during the current study are available from the corresponding author on reasonable request.

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

## Acknowledgements

This work was supported by the National Basic Research of China (Grants 2015CB932500), National Natural Science Foundations of China (Grant 51661135025, 51522207, 51572016, and U1530401). G.O. and K.H. acknowledge the support from Project funded by China Postdoctoral Science Foundation (2016M600079 and 2016M601019). We also thank Prof. H.X. Ju and Mr. S.W. Hu (Beamline BL10B of Hefei National Synchrotron Radiation Laboratory (NSRL) in China) for their great help in the XPS experiment. This research is supported by a Tianhe-2JK computing time award at the Beijing Computational Science Research Center (CSRC).

## Author contributions

G.O., Y.X., Y. Liang, and C.Y. fabricated the samples. G.O., Y.X., R.L., K.H., D.Z., and W. C. performed the measurements. B.G. and R.Y. performed the TEM results. B.W., Y.T., J. L. and L.-M.L. developed the theory. H.W. and Y. Li designed the project. All authors discussed the result, and contributed to the writing of the manuscript.

## Additional information

**Competing interests:** The authors declare no competing interests.

