## [Peer Review File · Nature Communications]

Reviewer #1 (Remarks to the Author):

The authors describe a novel method to tune defects presence in metal oxides. They also characterise in good detail the materials developed.

The topic is not new as well as the materials employed, however I found the paper interesting and exhaustive in general. I have a few comments to the authors here below :

The performance in the application could be an important aspect to evaluate the relevance of the work. The author should compare the photocatalytic hydrogen evolution rate that they achieved with the SoA.

The authors show higher defect density and, for the same material, higher conductivity, they give some explanation in the conclusions of the paper, but the transport mechanism should be explained better, as the material is mesoporous.

Reviewer #2 (Remarks to the Author):

Authors investigate a room temperature reduction process for binary oxides using Li metal and the property, especially photocatalytic property of TiO_{2-x}. The results show oxide powders (TiO₂, ZnO, SnO₂ and CeO₂) were reduced by room temperature grinding with Li metal powders in dimethyl carbonate. The obtained TiO_{2-x} exhibited high photocatalytic properties. This method seems a new facile way to reduce oxides, however, there is not so much sufficient impact. The metal reduction using Al at around 500 degC has been reported (high temperature but safe "ref: Energy Environ. Sci. 2013, 6, 3007"), and laser reduction has also been reported (not good for bulk materials but room temperature ambient atmosphere "ref: J. Mater. Chem. A 2014, 2, 6762"). The experimental procedure of Li metal method is very simple, however, it needs strictly managed environment to safely treat Li metal powders. Therefore, this method also has merit and demerit as well as the other methods.

I feel a critical problem about the manuscript itself. Most important point is mechanism of reduction using Li at room temperature just by grinding in dimethyl carbonate. However, it is not explained at all in the text. The analysis and characterization of the property for the obtained powder are no problem, and these information are not important (e.g. it has been already reported in so many papers that reduced TiO_{2-x} has several times higher photocatalytic property than pristine TiO₂). But there is hardly sufficient value, if authors makes the mechanism clear. Why is it possible to reduce oxides just by grinding with very small amount Li metal powders? Why does Li capture oxygen preferentially from oxides, not from organic solvent? Why is such small amount of Li powder enough for homogeneous reduction of oxide powders? How about the effect of Li metal powder size on the reduction? How does the amount of solvent influence the reduction? If the reduction of oxide proceed only at the contact interface with the Li powders, these points must be clear. If one Li particle has contact with an oxide particle and then the Li is simultaneously fully oxidized, it would be difficult to make homogeneously reduced oxide powders. Therefore, so many critical questions remain.

Thus, I do not recommend to publish this manuscript in Nature Communication.

Response Letter

Reviewer #1:

Comment 1: *The authors describe a novel method to tune defects presence in metal oxides. They also characterize in good detail the materials developed. The topic is not new as well as the materials employed, however I found the paper interesting and exhaustive in general.*

Response: We thank this reviewer for the positive evaluation of our work. We appreciate that this reviewer thinks our work “*interesting and exhaustive in general*”.

The reviewer concerns that “*The topic is not new as well as the materials employed*”.

In fact, we did not write clearly that the main task of this work is to tune defect engineering of oxides in the last version. Defect engineering has been considered as an effective strategy to tune the physical and chemical properties of materials and further broaden their applications stemmed from the versatile electronic properties of defective materials^{R1-R7}. As shown in this work, the defective oxides also exhibit high performance for photocatalytic hydrogen evolution, while they should have more potential applications. The main achievement of this work is that we realized the defect engineering of oxides at mild conditions by a simple approach.

Meanwhile, we fully agree that the oxide materials in our study (including TiO₂, ZnO, SnO₂ and CeO₂) have been intensely studied for many years as traditional material systems with a long history. Most of the recognitions on the oxides are confined to the stoichiometric oxides, while the nonstoichiometric oxides through defect engineering exhibit superior performance in many fields^{R1,R2,R6-R9}. While the challenging issues is

to control the defect effectively. In this work, the main task is to make the “old” oxides “sparkle” through the defect engineering.

In this article, we propose a lithium reduction strategy to controllably tune the defects in oxide materials at mild conditions with advantages of all-room-temperature processing, easiness to control, fastness, versatility and scalableness. Furthermore, the as-prepared defective TiO₂ nanopowders by lithium reduction strategy demonstrated superior photocatalytic performance. Indeed, we are not only reporting a new method to tune the defects in oxide materials, but also introducing new idea to solve the problems in defect engineering areas. Moreover, please note that the widely reported oxide materials in the past decades will not damage the novelty of this work; instead, the creation of lithium reduction strategy will greatly promote the final application of these oxide materials.

To summary, in this article, we propose a lithium reduction strategy to controllably tune the defects in oxide materials at mild conditions with many advantages. Till now, this defects engineering strategy has not yet been reported. Therefore, we firmly believe our results will attract lots of attention from science community.

Corresponding changes in page 2, line 6:

Our lithium reduction strategy to tune defects in oxide materials shows advantages like all-room-temperature processing, easiness to control, fastness, versatility and scalableness. As one of potential applications, the performance for photocatalytic hydrogen evolution of the defective TiO₂ was examined, with the hydrogen evolution rate being up to 41.8 mmol g⁻¹ h⁻¹ under one solar light irradiation, which is ~3 times

higher than pristine nanoparticles. The strategy of tuning the defect oxides used in this work should be beneficial for the many related applications.

Corresponding changes in page 3, line 1:

The ability to tune the defect structure of oxide materials has been a major area of focus since their fundamental physical and chemical properties greatly rely on their defect structures¹⁻⁷.

Corresponding changes in page 3, line 17:

Meanwhile, the defect engineering is an effective way to tune the electronic structure of metal oxides, which is vital for many applications^{2,5,23-26}.

Corresponding changes in page 9, line 11:

As discussed in the introduction, the defective oxides usually exhibits the superior properties, thus they exhibits good performance in many applications^{2,5,23-26}. Thus we expect that the defective metal oxides fabricated in this work should have many applications. In the following, we examined the photocatalytic degradation of organic pollutants and hydrogen evolution of TiO₂ nanopowders as potential applications.

Reference:

- R1. Koketsu, T. *et al.* Reversible magnesium and aluminium ions insertion in cation-deficient anatase TiO₂. *Nat. Mater.* **16**, 1142-1148 (2017).
- R2. Tan, H. *et al.* Efficient and stable solution-processed planar perovskite solar cells via contact passivation. *Science* **355**, 722-726 (2017).
- R3. Kim, H. S. *et al.* Oxygen vacancies enhance pseudocapacitive charge storage properties of MoO_{3-x}. *Nat. Mater.* 4810 (2016).
- R4. Yang, W. S. *et al.* Iodide management in formamidinium-lead-halide-based perovskite layers for efficient solar cells. *Science* **356**, 1376 (2017).

- R5. Giordano, F. *et al.* Enhanced electronic properties in mesoporous TiO₂ via lithium doping for high-efficiency perovskite solar cells. *Nat. Commun.* **7**, 10379 (2016).
- R6. Gong, C. *et al.* Discovery of intrinsic ferromagnetism in two-dimensional van der Waals crystals. *Nature* **546**, 265-269 (2017).
- R7. Yang, Y. *et al.* Deciphering chemical order/disorder and material properties at the single-atom level. *Nature* **542**, 75-79 (2017).
- R8. Sambur, J. B. *et al.* Sub-particle reaction and photocurrent mapping to optimize catalyst-modified photoanodes. *Nature* **530**, 77-80 (2016).
- R9. Zhang, X. *et al.* Direction-specific van der Waals attraction between rutile TiO₂ nanocrystals. *Science* **356**, 434-437 (2017).

Comment 2: *The performance in the application could be an important aspect to evaluate the relevance of the work. The author should compare the photocatalytic hydrogen evolution rate that they achieved with the SoA.*

Response: We thank this reviewer for the excellent suggestion. We have added them in the revised manuscript.

Corresponding changes in page 12, line 8:

Furthermore, we also compare the photocatalytic hydrogen evolution activity of the lithium reduced TiO₂ nanoparticles with the previous reported works (Table S2), indicating our defective TiO₂ by lithium reduction possesses superior photocatalytic activity.

Table S2. Comparison of photocatalytic hydrogen evolution activity with the available experiments

Catalyst	Hydrogen evolution rate	Light source	SoA	Refs
----------	-------------------------	--------------	-----	------

(mmol g ⁻¹ h ⁻¹)				
1% Au/TiO ₂	10.2	Iron halogenide mercury arc lamp, 250 W	6% CH ₃ OH	61
0.6% Pt/TiO ₂	10	AM 1.5 solar simulator	50% CH ₃ OH	18
1% PtO/TiO ₂	4.4	Xe lamp, 300 W	30% CH ₃ OH	62
1% Pt/TiO ₂	29	Xe lamp, 300 W	50% CH ₃ OH	63
1% Pt/TiO ₂	6.32	Xe arc lamp, 300 W	20% CH ₃ OH	64
1% Pt/TiO ₂	6.5	Xe lamp, 300 W	25% CH ₃ OH	65
1% Pt/TiO ₂	2.15	Xe arc lamp, 300 W	20% CH ₃ OH	66
0.5% Pt/TiO ₂	6.4	Hg lamp, 300 W	25% CH ₃ OH	20
1% Pt/TiO ₂	43.2	Xe lamp, 400 W	20% CH ₃ OH	19
1% Pt/TiO ₂	4	Xe lamp, 200 W	10% CH ₃ OH	67
0.5% Pt/TiO ₂	15	Xe lamp, 300 W	20% CH ₃ OH	31
1% Pt/TiO ₂	2	Xe lamp, 300 W	30% CH ₃ OH	68
0.57% Pd/TiO ₂	3.32	Xe lamp, 300 W	20% CH ₃ OH	8
1% Pt/TiO ₂	2.41	Xe lamp, 300 W	20% CH ₃ OH	69
0.5% Pt/TiO ₂	5.2	Xe lamp, 300 W	20% CH ₃ OH	70
1% Pt/TiO ₂	10.6	Xe lamp, 300 W	50% CH ₃ OH	71
0.5% Pt/TiO ₂	1.5	Xe lamp, 300 W	20% CH ₃ OH	72
1% Pt/TiO ₂	41.8	Xe lamp, 300 W	20% CH ₃ OH	This work

Reference:

61. Chiarello, G. L., Forni, L. & Selli, E. Photocatalytic hydrogen production by

- liquid- and gas-phase reforming of CH₃OH over flame-made TiO₂ and Au/TiO₂. *Catal. Today* **144**, 69-74 (2009).
62. Li, Y. H. *et al.* Unidirectional suppression of hydrogen oxidation on oxidized platinum clusters. *Nat. Commun.* **4**, 2500 (2013).
63. Cai, J. M. *et al.* In situ formation of disorder-engineered TiO₂(B)-anatase heterophase junction for enhanced photocatalytic hydrogen evolution. *ACS Appl. Mater. Interfaces* **7**, 24987-24992 (2015).
64. Tian, J., Leng, Y. H., Cui, H. Z. & Liu, H. Hydrogenated TiO₂ nanobelts as highly efficient photocatalytic organic dye degradation and hydrogen evolution photocatalyst. *J. Hazard. Mater.* **299**, 165-173 (2015).
65. Tan, H. Q. *et al.* A facile and versatile method for preparation of colored TiO₂ with enhanced solar-driven photocatalytic activity. *Nanoscale* **6**, 10216-10223 (2014).
66. Zheng, Z. K. *et al.* Hydrogenated titania: Synergy of surface modification and morphology improvement for enhanced photocatalytic activity. *Chem. Commun.* **48**, 5733-5735 (2012).
67. Li, L. *et al.* Sub-10 nm rutile titanium dioxide nanoparticles for efficient visible-light-driven photocatalytic hydrogen production. *Nat. Commun.* **6**, 5881 (2015).
68. Zhao, Z. *et al.* Effect of defects on photocatalytic activity of rutile TiO₂ nanorods. *Nano Res.* **12**, 4061-4071 (2015).
69. Hu, W. Y. *et al.* Facile strategy for controllable synthesis of stable mesoporous black TiO₂ hollow spheres with efficient solar-driven photocatalytic hydrogen evolution. *J. Mater. Chem. A* **4**, 7495-7502 (2016).
70. Zhu, G. L. *et al.* Black titania for superior photocatalytic hydrogen production and photoelectrochemical water splitting. *ChemCatChem* **7**, 2614-2619 (2015).
71. Wang, Y. T. *et al.* Hydrogenated cage-like titania hollow spherical photocatalysts for hydrogen evolution under simulated solar light irradiation. *ACS Appl. Mater. Inter.* **8**, 23006-23014 (2016).
72. Zhang, K. F. *et al.* Black N/H-TiO₂ nanoplates with a flower-like hierarchical

architecture for photocatalytic hydrogen evolution. *ChemSusChem* **9**, 2841-2848 (2016).

Comment 3: *The authors show higher defect density and, for the same material, higher conductivity, they give some explanation in the conclusions of the paper, but the transport mechanism should be explained better, as the material is mesoporous.*

Response: We thank this reviewer for the insight. As shown in many previous works, the existence of the Ti^{3+} can enhance the conductivity^{R10-R13}. Here, the electronic states for the reduced TiO_2 with different O_v concentrations were explored in the new version. The oxygen vacancy will induce the excess electrons in the n-type metal oxides. As shown in Figure S8, when the system contains the oxygen vacancy, the excess electrons are mainly localized on the Ti atoms, which make Ti^{4+} into Ti^{3+} . Meanwhile the concentration of the Ti^{3+} is improved with the increase of the O_v . It is worth noting that the excess electrons can freely diffuse with a small energy barrier of 0.3 eV^{R14}, confirmed by the first-principles molecular dynamics^{R15,R16}. Thus the main reason for the high conductivity should come from the excess electrons (Ti^{3+}) induced by the O_v .

Corresponding changes in page 7, line 7:

It is well-known that the existence of the Ti^{3+} can enhance the conductivity of the TiO_2 ³³⁻³⁵. As shown in Figure S8, the existence of oxygen vacancies induces the excess electrons, which make the Ti^{4+} into Ti^{3+} . Meanwhile the concentration of Ti^{3+} greatly depends on the amount of the O_v . The excess electrons can freely hop at the room

temperature³⁶⁻³⁸, which should be the main reason for the high conductivity of the reduced TiO₂.

Figure S8. The side view of defect TiO₂(101) with different subsurface O_v (blue circles) concentrations. Here, the 1O_v (a), 2O_v (b) and 4O_v (c) per formula were considered in our simulations, and the O_v stay in the subsurface. The spin density of the occupied states for different O_v concentrations is also shown in yellow and blue contours for the two different phases. The red and light blue spheres represent O and Ti atoms, respectively. O_vs are shown in the blue circles.

Reference:

- R10. Tang, K. C. *et al.* Distinguishing oxygen vacancy electromigration and conductive filament formation in TiO₂ resistance switching using liquid electrolyte contacts. *Nano Lett.* **17**, 4390-4399 (2017).
- R11. Lu, X. J. *et al.* Conducting interface in oxide homojunction: Understanding of superior properties in black TiO₂. *Nano Lett.* **16**, 5751-5755 (2016).
- R12. Amano, F., Nakata, M., Yamamoto, A. & Tanaka, T. Effect of Ti³⁺ ions and conduction band electrons on photocatalytic and photoelectrochemical activity of rutile titania for water oxidation. *J. Phys. Chem. C* **120**, 6467-6474 (2016).
- R13. Deskins, N. A. & Dupuis, M. Electron transport via polaron hopping in bulk TiO₂: A density functional theory characterization. *Phys. Rev. B* **75**, 19521219 (2007).
- R14. Kowalski, P. M., Camellone, M. F., Nair, N. N., Meyer, B. & Marx, D. Charge

localization dynamics induced by oxygen vacancies on the titania $\text{TiO}_2(110)$ surface. *Phys. Rev. Lett.* **105**, 146405 (2010).

- R15. Spreafico, C. & VandeVondele, J. The nature of excess electrons in anatase and rutile from hybrid DFT and RPA. *Phys. Chem. Chem. Phys.* **16**, 26144-26152 (2014).

Reviewer #2:

Comment 1: *Authors investigate a room temperature reduction process for binary oxides using Li metal and the property, especially photocatalytic property of TiO_{2-x} . The results show oxide powders (TiO_2 , ZnO , SnO_2 and CeO_2) were reduced by room temperature grinding with Li metal powders in dimethyl carbonate. The obtained TiO_{2-x} exhibited high photocatalytic properties. This method seems a new facile way to reduce oxides.*

Response: We thank this reviewer and feel grateful that the reviewer has carefully read our manuscript.

Comment 2: *However, there is not so much sufficient impact. The metal reduction using Al at around 500 degC has been reported (high temperature but safe "ref: Energy Environ. Sci. 2013, 6, 3007"), and laser reduction has also been reported (not good for bulk materials but room temperature ambient atmosphere "ref: J. Mater. Chem. A 2014, 2, 6762").*

Response: We thank the reviewer for pointing out these important references from Prof. Huang's group and Prof. Nakajima's group. We apologize for not making the novelty of this study clear enough in our manuscript and would like to give more discussion on our scientific contribution. We strongly agree with this reviewer that defects engineering have become a very hot and important topic in science community, and some recent breakthroughs on the defects engineering of TiO_2 materials have been recently reported, for example, Nakajima *et al.* reported defective

TiO₂ material by high energy laser irradiation, which provides us very important strategy to tune the defects in oxide materials^{R17,R18}. In addition, chemical reduction route to generate defects in TiO₂ has also been reported as an effective strategy, like high temperature reduction by H₂, Mg, Al and other reductive chemicals, however, these reduction often required high-temperature, pressure control and/or long-term processing (Table R1), which usually lead to crystal structural and morphological changes of nanomaterials. Furthermore, it is difficult to control the concentration of defects by such high temperature reduction. We compared our reduction method with previous reported methods in Table R1 as attached below.

To summary, we not only reported defective TiO₂ photocatalyst with high activity and stability, but also developed new strategy for controllable preparation of defective oxides with remarkable advantages that have not been reported before in other works.

The Li reduction strategy in this report significantly showed following advantages:

1. All-room-temperature processing: Most of the oxides can be reduced by Li metal at room-temperature due to its high reductive activity; meanwhile, we can engineer the defects in oxides without changing their morphology and crystal structure at room-temperature.
2. Easiness to control: It is one of the most promising but challenging question to easily tune the content of defects in oxides in the field of defects engineering. Any breakthrough could contribute not only to the photocatalysts, but to the whole field of defect engineering. Although great effect has been made, as far as we know, it is still a big challenge for this field. In our lithium reduction strategy, we

have successfully tuned the content of defects in oxides by simply adjusting the weight ratio of lithium powder to oxides, showing significant advantages than the previous reports (Figure R1).

3. Fastness, versatility and scalableness: As demonstrated in our manuscript, a series of defective oxides have been prepared by simply mixing and grinding the target oxide with lithium powder in a glove box in a few hours without expensive equipment. Meanwhile, the amount of the defective oxides can be easily scaled up as the commercialized lithium metal powder has been supplied in large quantities.

Therefore, we believe this work opens up a new methodology different from previous ones for defects engineering of oxide materials, which will have profound influence on future defective oxides research.

We added more supporting information:

Table R1. Technical comparison of lithium reduction strategy with the previous reported methods

Method	Conditions	Equipment	Catalyst	Refs.
Aluminium reduction	0.5 Pa, Al@800 °C and TiO ₂ @500 °C, 6 h	A two-zone tube furnace	TiO ₂ nanopowders	R16
Pulsed ultraviolet laser irradiation	500 Pa, Laser irradiation, 150 s	Pulsed KrF laser	TiO ₂ thin film	R17
Lithium reduction	Room-temperature, Atmospheric Pressure, 1 h	Glove box	TiO ₂ nanopowders	This work

Corresponding changes in page 3, line 12:

Reductive metals have also been introduced to tune the defects in TiO₂ material at high temperatures, such Mg and Al^{19,20}. In addition, Nakajima *et al.* reported defective TiO₂ material by high energy laser irradiation, which provides us another strategy to tune the defects in oxide materials^{21,22}. Nevertheless, it keeps challenging to controllably engineer the defects in oxide materials at ambient conditions. Meanwhile, the defect engineering is an effective way to tune the electronic structure of metal oxides, which is vital for many applications^{2,5,23-26}.

Corresponding changes in page 15, line 5:

We developed a lithium reduction approach to conveniently implant defects in oxide materials with advantages of all-room-temperature processing, easiness to control, fastness, versatility and scalableness. Besides, the defects in TiO₂ nanoparticles were successfully tuned, which demonstrated significantly enhanced photocatalytic activity and stability in degradation and hydrogen evolution reactions after lithium reduction treatment. In addition, a series of oxide materials, including ZnO, SnO₂ and CeO₂, have also been successfully modified through implanted defects by a similar Li reduction process. Moreover, it is firmly believed that the properties of other functional oxide materials can be optimized by the proposed lithium reduction approach, which can be applied to many areas such as environmental protection, microelectronics, energy conversion and storage.

Figure R1. Photograph of oxide nanopowders reduced by different weight ratio of lithium to oxides. **a**, TiO₂. **b**, ZnO. **c**, SnO₂. **d**, CeO₂.

Reference:

- R16. Wang, Z. *et al.* Visible-light photocatalytic, solar thermal and photoelectrochemical properties of aluminium-reduced black titania. *Energy Environ. Sci.* **6**, 3007-3014 (2013).
- R17. Nakajima, T., Nakamura, T., Shinoda, K. & Tsuchiya, T. Rapid formation of black titania photoanodes: Pulsed laser-induced oxygen release and enhanced solar water splitting efficiency. *J. Mater. Chem. A* **2**, 6762-6771 (2014).
- R18. Nakajima, T., Tsuchiya, T. & Kumagai, T. Pulsed laser-induced oxygen deficiency at TiO₂ surface: Anomalous structure and electrical transport properties. *J. Solid State Chem.* **182**, 2560-2565 (2009).

Comment 3: *The experimental procedure of Li metal method is very simple, however, it*

needs strictly managed environment to safely treat Li metal powders. Therefore, this method also has merit and demerit as well as the other methods.

Response: We thank this reviewer for the valuable comments. The lithium reduction of materials has been widely applied not only for scientific study but also in industry production. For example, in industrial, Li metal is widely used as a flux for welding or soldering metals to promote the fusing of metal during the process, eliminates the forming of oxides by absorbing impurities^{R19}. In addition, Li metal also has important applications for energy storage^{R20}.

We agree that Li metal should be strictly managed due to its high activity, while it only requires a glove box in our experiment. Firstly, the lithium powders used in this work were massively produced in industry and treated by surface passivation, showing a high degree of security. Besides, our lithium powders were put into a sealed glove box full of argon for the safe storage and usage. Lastly, just a small amount of lithium powders (~3 wt%) were required in our method, meanwhile, the reaction product is a non-hazardous lithium oxides which can be easily removed by washing with water. In order to clearly show the safety and advance of our lithium reduction strategy, the detailed process for the preparation of defective TiO₂ nanopowders reduced by 5 wt% Li was recorded and revealed in Figure R2.

We gave more discussion on this point. Corresponding changes in page 26, line 13:

It is noted that although highly active Li metal powders were used in the experiment, there are no danger in the whole experimental process in regular operation. Firstly, the lithium powders used in this work were massively produced in industry and treated by

surface passivation, showing a high degree of security. Besides, the lithium powers were put into a sealed glove box full of argon for the safe storage and usage. Lastly, just a small amount of lithium powders (less than 5 wt%) were required by our method.

Figure R2. Photograph of detailed process for the preparation of defective TiO₂ nanopowders reduced by 5 wt% Li.

Reference:

- R19. Garrett, D. E. Handbook of lithium and natural calcium chloride. *Academic Press*, 2004, p200.
- R20. Wadia, C., Albertus, P., & Srinivasan, V. Resource constraints on the battery energy storage potential for grid and transportation applications. *J. Power*

Sources. **196**, 1593-1598 (2011).

Comment 4: *I feel a critical problem about the manuscript itself. Most important point is mechanism of reduction using Li at room temperature just by grinding in dimethyl carbonate. However, it is not explained at all in the text.*

Response: We thank this viewer for his/her critical comments. We would like to apologize for not making our mechanism clear enough in our manuscript, which might cause some confusion on the understanding of our work. The detailed mechanism has been added in our revised manuscript. Firstly, the ΔG for reaction $4\text{Li} + \text{TiO}_2 \rightarrow \text{Ti} + 2\text{Li}_2\text{O}$ is $-233.6 \text{ kJ mol}^{-1}$ (or 2.43 eV/atom) at 298 K, indicating the reaction can occur spontaneously at room-temperature. Thus the lithium metal can draw out O from TiO_2 to generate Li_2O as well as defective TiO_2 in the grinding process. Secondly, the lithium was continuously extended during the grinding process because of the high ductility, resulting in the constant exposure of fresh lithium to react with the oxides, so that the lithium metal would react fully. Lastly, the purpose of adding dimethyl carbonate is to allow the lithium metal to be in fully contact with the oxides during the experiment. For a detailed explanation, please see:

Corresponding changes in page 4, line 9:

For example, the ΔG for reaction $4\text{Li} + \text{TiO}_2 \rightarrow \text{Ti} + 2\text{Li}_2\text{O}$ is $-233.6 \text{ kJ mol}^{-1}$ at 298 K, indicating the reaction can occur spontaneously at room-temperature.

Corresponding changes in page 8, line 14:

From the above results, we can see that uniform defects have been successfully implanted at the surface of the oxide nanopowders by our lithium reduction strategy, which can be ascribed to the following three reasons. Firstly, the lithium metal can act with the oxides at room-temperature according to the Ellingham diagrams and our calculated results. Thus the lithium metal can draw out O from the oxides to generate Li_2O as well as defective oxides in the grinding process. Secondly, the lithium can be continuously extended during the grinding process because of its high ductility, resulting in the constant exposure of fresh lithium to react with the oxides, so that the lithium metal would react fully. Lastly, the lithium powders and oxide nanopowders can be uniformly mixed with the adding of dimethyl carbonate, which is beneficial for the uniform implantation of defects at the surface of the oxide nanopowders in the grinding process.

Comment 5: *The analysis and characterization of the property for the obtained powder are no problem, and these information are not important (e.g. it has been already reported in so many papers that reduced TiO_{2-x} has several times higher photocatalytic property than pristine TiO_2). But there is hardly sufficient value, if authors make the mechanism clear.*

Response: We thank the reviewer for this suggestion. We have added the detailed mechanism in the revised manuscript.

Corresponding changes in page 14, line 11:

From the above results, it can be clearly seen that the lithium reduced TiO_2

nanoparticles show significantly enhanced photocatalytic property compared with the pristine TiO₂, which may be correlated with the greatly increased light absorption, improved conductivity, surface disorder layer, implanted oxygen vacancies and generated Ti³⁺. In particular, the improved light absorption and photocatalytic properties are attributed to the Ti³⁺ of the defective TiO₂. Furthermore, metallic conduction can be achieved at the crystalline-amorphous interface of the defective TiO₂ nanoparticles, which would enhance the electron transport property of TiO₂³⁴. Another important factor is the implanted oxygen vacancies and/or surface disorder, which also plays a great role in increasing the photocatalytic activity. The donor density of TiO₂ is enhanced by introducing oxygen vacancies as they can act as electron donors⁴⁹, which would improve charge transport and shift the Fermi level toward conduction band⁵⁰, facilitating charge separation and improving the IPCE in UV region⁵¹. Finally, the generated Ti³⁺ in the defective TiO₂ can reduce the recombination of the photogenerated electron-hole pairs and thus improve the photocatalytic activity of the lithium reduced TiO₂ nanoparticles¹⁹.

Comment 6: *Why is it possible to reduce oxides just by grinding with very small amount Li metal powders?*

Response: We thank the reviewer for the comment. We have checked many times in our experiments that only small amount of Li is required to obtain the defects. This can be explained by considering the thermal dynamics and kinetics of the lithium reduction reactions.

First of all, the reduction has enough driving force at room temperature. The ΔG for

reaction $4\text{Li}+\text{TiO}_2\rightarrow\text{Ti}+2\text{Li}_2\text{O}$ is -233.6 KJ/mol (or 2.43 eV/atom), indicating the reaction can occur spontaneously at room-temperature. Furthermore, the calculated reaction enthalpy for $x\text{Li}+\text{TiO}_2\rightarrow\text{TiO}_{2-x/2}+x/2\text{Li}_2\text{O}$ is -4.03 eV (Table S1), where the negative value means an exothermic process. Thus lithium metal can capture O from TiO_2 to generate Li_2O as well as defective TiO_2 in the grinding process. In addition, the relative atomic mass of lithium is relatively small (6.941). According to reaction equation $x\text{Li}+\text{TiO}_2\rightarrow\text{TiO}_{2-x/2}+x/2\text{Li}_2\text{O}$, when 5 wt% Li was added to TiO_2 , the theoretical product was $\text{TiO}_{1.712}$ after completion of the reaction. In other words, the content of oxygen defect in the final product was 14.4% . So a small amount of lithium metal powder can be effective in reducing the oxides to produce defective oxides. Lastly, the lithium atoms have high diffusion speed in oxide materials; also benefit the controlled formation of oxygen defects.

We give more discussion on this point. Corresponding changes in page 8, line 14:

From the above results, we can see that uniform defects have been successfully implanted at the surface of the oxide nanopowders by our lithium reduction strategy, which can be ascribed to several reasons. Firstly, the lithium metal can act with the oxides at room-temperature according to the Ellingham diagrams and our calculated results. Thus the lithium metal can draw out O from the oxides to generate Li_2O as well as defective oxides in the grinding process. Secondly, the lithium can be continuously extended during the grinding process because of its high ductility, resulting in the constant exposure of fresh lithium to react with the oxides, so that the lithium metal would react fully. Lastly, the lithium powders and oxide nanopowders

can be uniformly mixed with the adding of dimethyl carbonate, which is beneficial for the uniform implantation of defects at the surface of the oxide nanopowders in the grinding process.

Comment 7: *Why does Li capture oxygen preferentially from oxides, not from organic solvent?*

Response: We thank the reviewer for the comment. In fact, the solvents have high chemical stability with Lithium metal and have been applied widely as components of electrolytes in Li-ion batteries. First of all, based on the above thermodynamic calculations, lithium metal can capture O from TiO_2 to generate Li_2O and defective TiO_2 at room-temperature. Our first-principles calculations show that the formation energy of the oxygen vacancy in TiO_2 is 3.77 eV per oxygen, while the corresponding formation energy of losing one oxygen in DMC is 4.80 eV per oxygen, which is 1.03 eV larger than the corresponding one through capturing the oxygen from TiO_2 . Such results clearly suggest the reduction of DMC is more difficult to occur compared with TiO_2 , and the Li prefers to capture O from the TiO_2 instead of organic solvent. At last, DMC as a dispersant is widely used in lithium-ion battery areas as an important component part of the organic electrolyte^{R21,R22}, which means this solvent has good stability with lithium metal at room-temperature.

Figure R3. The model used for reduction of dimethyl carbonate (DMC). Left is the perfect DMC, and the right one is the reduced DMC. The brown, red and white spheres stand for C, O and H atoms, respectively.

Corresponding changes in page 8, line 9:

Our first-principles calculations show that the formation energy of the oxygen vacancy in TiO_2 is 3.77 eV per oxygen, while the corresponding formation energy of losing one oxygen atom in DMC is 4.80 eV per oxygen, which is 1.03 eV larger than the corresponding one through capturing the oxygen from TiO_2 . Thus the Li prefers to capture oxygen from the TiO_2 instead of the organic molecules.

Reference:

- R21. Rodrigues, M. T. F. et al. A materials perspective on Li-ion batteries at extreme temperatures. *Nature Energy* **2**, 17108 (2017).
- R22. Song, H. et al. Anomalous decrease in structural disorder due to charge redistribution in Cr-doped $\text{Li}_4\text{Ti}_5\text{O}_{12}$ negative-electrode materials for high-rate Li-ion batteries. *Energy Environ. Sci.* **5**, 9903-9913 (2012).

Comment 8: *Why is such small amount of Li powder enough for homogeneous reduction of oxide powders?*

Response: We thank the reviewer for the comment. Firstly, the lithium metal was continuously extended during the grinding process because of its high ductility. This led to the constant exposure of fresh lithium to react with the oxide, so that lithium would react fully and generate defective oxide. Secondly, our oxide materials are approximately spherical and they can scroll in the grinding process, so the surface of

the nanopowders has an equal opportunity to react with the lithium to form an oxide material with evenly surface defects distribution. Furthermore, as shown in Figure 2e, defects are mainly distributed at the surface regions of the defective oxide nanopowders. Thirdly, although the oxygen vacancy is produced in the surface region, it is well-known that O_v in anatase prefers to diffuse from the surface to the subsurface and deep layers^{R23-R25}. As shown in the Figure S22, our first-principles calculations suggested that the O_v initially created on the surface can easily diffuse from the surface to the subsurface, as reported in previous experiment and theory^{R23-R25}. Then, the surface O_v could diffuses from the surface into the subsurface with a small diffusion barrier (~ 0.7 eV), and the relatively small energy barrier could help the homogeneous distribution of O_v .

Corresponding changes in page 9, line 3:

In addition, the reaction between the Li and TiO_2 occur at the interface region, which captures the oxygen atoms of the TiO_2 surface. Although the oxygen vacancies are initially generated at the surface regions, they prefer to diffuse into the subsurface or the inner layer (Figure S22). The subsurface O_v is more stable by about 0.13 eV than the one in surface, as suggested in the previous experiment and theory³⁹⁻⁴¹. The calculated energy barrier is about 0.7 eV for the diffusion barrier (Figure S22), which indicates that the oxygen vacancy has the tendency to diffuse from the surface to the inner layer region at the room temperature.

Figure S22. The reaction pathway of the O_v diffusion from surface (the left one) to subsurface (the right one). The total energy of surface O_v is set as the zero energy for reference. The light blue and red balls are Ti atoms and O atoms, respectively. The yellow ball stands for the moving oxygen atom during the O_v diffusion. The blue circle denotes the O_v defect. The transition states are shown on the up panel and the stable O_v sites are shown on the down panel.

Reference:

- R23. Cheng, H. Z. & Selloni, A. Energetics and diffusion of intrinsic surface and subsurface defects on anatase $TiO_2(101)$. *J. Chem. Phys.* **131**, 54703 (2009).
- R24. He, Y. B., Dulub, O., Cheng, H. Z., Selloni, A. & Diebold, U. Evidence for the predominance of subsurface defects on reduced anatase $TiO_2(101)$. *Phys. Rev. Lett.* **102**, 10610510 (2009).
- R25. Cheng, H. Z. & Selloni, A. Surface and subsurface oxygen vacancies in anatase TiO_2 and differences with rutile. *Phys. Rev. B* **79**, 0921019 (2009).

Comment 9: *How about the effect of Li metal powder size on the reduction?*

Response: We thank the reviewer for the very helpful suggestion. We believe that the size of lithium powder has little effect on the reduction of oxides. Commercial lithium powder was used for our experiment with diameter of about 30 μm (Figure S11). Unfortunately, we have been looking for a long time but still haven't bought other sizes of lithium powder. In order to investigate the size effect of lithium on the reduction, we selected lithium powder and lithium piece as raw materials. Figure S12 shows the TiO_2 powder after grinding with 5 wt% Li powder (left) and Li piece (right). It is obvious that the defective TiO_2 produced with Li powder or Li piece exhibited the same color depth, indicating their defective content is consistent. In summary, we believe that the size of lithium powder has no effect on the reduction of oxides. In fact, the most important factor for this reaction is the content of lithium.

Corresponding changes in page 8, line 1:

In order to investigate the effect of Li metal powder size on the reduction of oxides, we prepared defective TiO_2 nanopowders deduced by 5 wt% Li powders and Li piece. Here, the diameter of the Li powders we used in the experiment is about 30 μm (Figure S11). As shown in Figure S12, it is apparent that the defective TiO_2 produced with Li powder or Li piece exhibited the same color depth, indicating their defective content is consistent.

Figure S11. SEM micrograph of lithium powders.

Figure S12. Photograph of TiO₂ nanopowders reduced by 5 wt% Li powders and Li piece respectively.

Comment 10: *How does the amount of solvent influence the reduction?*

Response: We thank the reviewer for the suggestion. Again, we conformed that the solvents have not join the reaction and mainly function to disperse the particles. In response to this comments, we performed new experiments to tune the amount of solvent added in the experiment. As shown in Figure S13, the weight ratio of oxides

to DMC is 1:10, 1:20 and 1:30, respectively. It can be seen that no significant changes can be observed on the color of the products as the solvent content changes.

Corresponding changes in page 8, line 6:

In addition, we also compared different content of dispersant (DMC) on the reduction of TiO_2 nanopowders. As shown in Figure S13, it can be seen that the color of the defective TiO_2 did not change with the dispersant content change from 1:10, 1:20 to 1:30.

Figure S13. Photograph of TiO_2 nanopowders reduced by 5 wt% Li powders with different weight ratio of oxides to DMC.

Comment 11: *If the reduction of oxide proceed only at the contact interface with the Li powders, these points must be clear. If one Li particle has contact with an oxide particle and then the Li is simultaneously fully oxidized, it would be difficult to make homogeneously reduced oxide powders.*

Response: We thank the reviewer for the very helpful comment. First of all, as the response in the Comment 8 rose by the same referee, although oxygen vacancies are

initially formed on the anatase TiO_2 surface, they are energetically stabilized on the subsurface and the inner layer^{R24}. This is the main reason why the oxygen vacancy on anatase surface is seldom observed while widely exist in rutile^{R25}. Our first-principles calculations show the diffusion barrier of oxygen vacancy from the surface to the subsurface is about 0.7 eV (see Figure S22), which agrees well with the previous works^{R23,R25}. Therefore, the initially created O_v on the anatase surface will easily diffuse into the subsurface because of the lower barrier for the diffusion. Consequently, the O_v will concentrate on the subsurface homogeneously.

There are several other reasons accounting for it. First, lithium was a reducing agent taking O from the oxide to produce a defective oxide material in our method. Especially, DMC was added as a dispersant in the experiment to allow sufficient contact between the lithium and the oxide to ensure uniformity of the reaction. Second, theoretically, lithium reacts with the oxide generating lithium oxide, which may hinder further reduction; however, in practice, the lithium was continuously extended during the grinding process because of the high ductility. This led to the constant exposure of fresh lithium to react with the oxide, so that lithium would react fully. Third, our oxide material is approximately spherical, so the surface of the nanopowders has an equal opportunity to react with the lithium to form an oxide material with evenly surface defects distribution.

Corresponding changes in page 9, line 3:

In addition, the reaction between the Li and TiO_2 occur at the interface region, which captures the oxygen atoms of the TiO_2 surface. Although the oxygen vacancies are

initially generated at the surface regions, they prefer to diffuse into the subsurface or the inner layer (Figure S22). The subsurface O_v is more stable by about 0.13 eV than the one in surface, as suggested in the previous experiment and theory³⁹⁻⁴¹. The calculated energy barrier is about 0.7 eV for the diffusion barrier (Figure S22), which indicates that the oxygen vacancy has the tendency to diffuse from the surface to the inner layer region at the room temperature.

Figure S22. The reaction pathway of the O_v diffusion from surface (the left one) to subsurface (the right one). The total energy of surface O_v is set as the zero energy for reference. The light blue and red balls are Ti atoms and O atoms, respectively. The yellow ball stands for the moving oxygen atom during the O_v diffusion. The blue circle denotes the O_v defect. The transition states are shown on the up panel and the stable O_v sites are shown on the down panel.

Reference:

- R23. Cheng, H. Z. & Selloni, A. Energetics and diffusion of intrinsic surface and subsurface defects on anatase $\text{TiO}_2(101)$. *J. Chem. Phys.* **131**, 54703 (2009).
- R24. He, Y. B., Dulub, O., Cheng, H. Z., Selloni, A. & Diebold, U. Evidence for the predominance of subsurface defects on reduced anatase $\text{TiO}_2(101)$. *Phys. Rev. Lett.* **102**, 10610510 (2009).
- R25. Cheng, H. Z. & Selloni, A. Surface and subsurface oxygen vacancies in anatase TiO_2 and differences with rutile. *Phys. Rev. B* **79**, 0921019 (2009).

Reviewer #1 (Remarks to the Author):

The authors have clarified the points of concern from my point of view, I therefore recommend for publication.

Reviewer #2 (Remarks to the Author):

The authors modified the manuscript corresponding to my concerns one by one, and the manuscript was much improved. But I request more evaluations for the obtained reduced particles before the publication.

1. Only the local structural information can be understood from the TEM data, indicating the defective structure. How about the averaged structure? It can be observed from some diffuse scattering of XRD or some other analyses? Readers do want to know this powder is not the mixture of TiO₂ and TiO_{2-x} but the homogeneous TiO_{2-x}. Even from the XRD data, it can be discussed.

2. How do you carry out the grinding? By hand? The details must be written in the Methods section. This is important information in this manuscript.

3. The contact reaction between the Li and TiO₂ powders during the grinding (by hand?) means that this reaction proceeded quite fast time scale. And then the oxygen diffusion in the TiO_{2-x} just after the capturing oxygen by the Li is also very smooth. Are these really reasonable? I'm not sure how many times one TiO₂ particle has chances to contact with the fresh Li powders. This point is very critical point in this reaction process.

Response Letter

Reviewer #1:

Comment 1: *The authors have clarified the points of concern from my point of view, I therefore recommend for publication.*

Response: We thank this reviewer for suggesting acceptance of our work.

Reviewer #2:

Comment 1: *Only the local structural information can be understood from the TEM data, indicating the defective structure. How about the averaged structure? It can be observed from some diffuse scattering of XRD or some other analyses? Readers do want to know this powder is not the mixture of TiO_2 and TiO_{2-x} but the homogeneous TiO_{2-x} . Even from the XRD data, it can be discusses.*

Response: We thank this reviewer for the comments. We agree with this reviewer that the TEM images can only exhibit local structures of single TiO_2 nanopowder, this is the nature of many materials characterization methods like TEM, SEM and AFM. However, such characterizations still provide important information since the particles observed in TEM were chosen randomly and we confirmed the structure on multiple samples. For XRD analysis, it is very difficult to determine whether lithium reduced TiO_2 nanopowders is *homogeneous TiO_{2-x}* or *mixture of TiO_2 and TiO_{2-x}* , since TiO_{2-x} material doesn't provide new XRD peaks, and the results are limited by the accuracy of XRD analysis. We can speculate that the lithium reduced TiO_2 nanopowders possess defective structure according to the following reasons: (1) Besides TEM, the

defect structures have also been confirmed by XPS spectra and EPR spectra (Figure 2 and S10), which provided the average analysis of the materials; (2) The defective oxide powders have demonstrated significantly enhanced photocatalytic property after lithium reduction. In addition, the materials experienced uniform color changes from white to black; (3) Our experimental procedure ensures that all substrates are evenly involved in the reaction. First, we chose oxide nanoparticles and Li powders instead of micro oxide powders and Li foil as the substrates to avoid the possible uneven contact. Second, we added solvent (DMC) to enhance their contact while grinding. Last, we applied long time (~1 h) grinding process to allow the oxide particles evenly attach and react with the lithium powders. (4) Our experimental and theoretical discussion also pointed out that the Li atoms have a high reaction rate with oxides; that is to say, the defect implanting reaction is fast in dynamic, which further assistant the formation of uniform defects in the nanoparticles.

Considering the significant improved material performance, we believed that on average, there are rich amount of active defects been implanted in the materials, and our future work will focus on developing more advanced grinding methods like machine-grinding or ball-milling in order to make more uniform materials. To address this comment, we add more discussion in the revised manuscript.

Corresponding changes in page 15, line 22:

Thus, we can speculate that almost all of our lithium reduced oxide nanopowders have defective structure based on the following reasons: (1) Besides TEM, the defect structures have also been confirmed by XPS spectra and EPR spectra (Figure 2 and

S10), which provided the average analysis of the materials; (2) The defective oxide powders have demonstrated significantly enhanced photocatalytic property after lithium reduction. In addition, the materials experienced uniform color changes from white to black; (3) Our experimental procedure ensures that all substrates are evenly involved in the reaction. First, we chose oxide nanoparticles and Li powders instead of micro oxide powders and Li foil as the substrates to avoid the possible uneven contact. Second, we added solvent (DMC) to enhance their contact while grinding. Last, we applied long time (~1 h) grinding process to allow the oxide particles evenly attach and react with the lithium powders. (4) Our experimental and theoretical discussion also pointed out that the Li atoms have a high reaction rate with oxides; that is to say, the defect implanting reaction is fast in dynamic, which further assistant the formation of uniform defects in the nanoparticles.

Comment 2: *How do you carry out the grinding? By hand? The details must be written in the Method section. This is important information in this manuscript.*

Response: We thank the reviewer for this comment. We did the powder grinding by hand in our experimental process, which has been added in the revised manuscript.

We give more details and discussions to address this comment.

Corresponding changes in page 28, line 8:

The materials were carefully mixed by hand grinding with speed of about 2 laps per second for 1 h and took out to dissolve the generated lithium oxide by dilute HCl.

Comment 3: *The contact reaction between the Li and TiO₂ powders during the grinding (by hand?) means that this reaction proceeds quite fast time scale. And then the oxygen diffusion in the TiO_{2-x} just after the capturing oxygen by the Li is also very smooth. Are these really reasonable? I'm not sure how many times one TiO₂ particle has chances to contact with the fresh Li powders. This point is very critical point in this reaction process.*

Response: We thank the reviewer for this comment. This valuable comment inspired us to a deeper thinking of the reaction mechanisms. We agree that the contact reaction between the Li and TiO₂ powders proceeds very fast in principle, since Li can diffuse fast in the oxides. However, in our practical experiments, the process becomes much slower since lithium oxide passivation layers (like Li₂O) formed at the interface between Li and defective oxide powders, buffering the reaction between Li and oxides. Therefore, only a small amount of Li diffused into oxides while Li and oxide particle surface contacted. However, with the hand grinding continuous, the oxide passivation layers was destroyed by the shearing and friction forces between particles, with fresh lithium exposed to react with fresh oxides surfaces. In long-time grinding process, the lithium oxide passivation layer was cyclically formed and destroyed, and the lithium finally fully reacted with oxide powders, resulting in the formation of uniform defective oxide powders. We observed from Figure S1 that the color of the mixed powders tuned darker after long time grinding, indicating that the defects formed with long time reactions. In our future study, ball milling or machine grinding can be introduced to speed up the reactions, making the process more controllable and

efficient. In order to make this point clear, we have added more discussion in the revised manuscript.

Corresponding changes in page 15, line 7:

We note that the contact reaction between the Li and TiO₂ powders proceeds very fast in principle, since Li can diffuse fast in the oxides. However, we observed from Figure S1 that the color of the mixed powders turned darker after long time grinding, indicating that the defects formed with long time reactions. This is because lithium oxide passivation layers (like Li₂O) formed at the interface between Li and defective oxide powders, buffering the reaction between Li and oxides. Therefore, only a small amount of Li diffused into oxides while Li and oxide particle surface contacted. However, with the hand grinding continuous, the oxide passivation layers was destroyed by the shearing and friction forces between particles, with fresh lithium exposed to react with fresh oxides surfaces. In long-time grinding process, the lithium oxide passivation layer was cyclically formed and destroyed, and the lithium finally fully reacted with oxide powders, resulting in the formation of uniform defective oxide powders. While we herein use hand grinding to demonstrate the successful Li-reduction and defect-implanting chemistry, ball milling or machine grinding can be introduced to speed up the reactions, making the process more controllable and efficient in the future study.

Corresponding changes in page 33, line 2:

Figure S1. Photograph of detailed process for the preparation of defective TiO₂ nanopowders reduced by 5 wt% Li.

Reviewer #2 (Remarks to the Author):

The manuscript has been improved by responding comments. This methode will give a new important progress to make oxygen deficient materials, thus, I recommend to publish this as the current form.

Reviewers' comments:

Reviewer #1 (Remarks to the Author):

The authors describe a novel method to tune defects presence in metal oxides. They also characterize in good detail the materials developed. The topic is not new as well as the materials employed, however I found the paper interesting and exhaustive in general. I have a few comments to the authors here below:

The performance in the application could be an important aspect to evaluate the relevance of the work. The author should compare the photocatalytic hydrogen evolution rate that they achieved with the SoA.

The authors show higher defect density and, for the same material, higher conductivity, they give some explanation in the conclusions of the paper, but the transport mechanism should be explained better, as the material is mesoporous.

Reviewer #2 (Remarks to the Author):

Authors investigate a room temperature reduction process for binary oxides using Li metal and the property, especially photocatalytic property of TiO_{2-x} . The results show oxide powders (TiO_2 , ZnO , SnO_2 and CeO_2) were reduced by room temperature grinding with Li metal powders in dimethyl carbonate. The obtained TiO_{2-x} exhibited high photocatalytic properties. This method seems a new facile way to reduce oxides, however, there is not so much sufficient impact. The metal reduction using Al at around 500 degC has been reported (high temperature but safe "ref: Energy Environ. Sci. 2013, 6, 3007"), and laser reduction has also been reported (not good for bulk materials but room temperature ambient atmosphere "ref: J. Mater. Chem. A 2014, 2, 6762"). The experimental procedure of Li metal method is very simple, however, it needs strictly managed environment to safely treat Li metal powders. Therefore, this method also has merit and demerit as well as the other methods.

I feel a critical problem about the manuscript itself. Most important point is mechanism of reduction using Li at room temperature just by grinding in dimethyl carbonate. However, it is not explained at all in the text. The analysis and characterization of the property for the obtained powder are no problem, and these information are not important (e.g. it has been already reported in so many papers that reduced TiO_{2-x} has several times higher photocatalytic property than pristine TiO_2).

But there is hardly sufficient value, if authors make the mechanism clear. Why is it possible to reduce oxides just by grinding with very small amount Li metal powders? Why does Li capture oxygen preferentially from oxides, not from organic solvent? Why is such small amount of Li powder enough for homogeneous reduction of oxide powders? How about the effect of Li metal powder size on the reduction? How does the amount of solvent influence the reduction? If the reduction of oxide proceed only at the contact interface with the Li powders, these points must be clear. If one Li particle has contact with an oxide particle and then the Li is simultaneously fully oxidized, it would be difficult to make homogeneously reduced oxide powders. Therefore, so many critical questions remain.

Thus, I do not recommend to publish this manuscript in Nature Communication.

Response Letter

Reviewer #1:

Comment 1: *The authors describe a novel method to tune defects presence in metal oxides. They also characterize in good detail the materials developed. The topic is not new as well as the materials employed, however I found the paper interesting and exhaustive in general.*

Response: We thank this reviewer for the positive evaluation of our work. We appreciate that this reviewer thinks our work “*interesting and exhaustive in general*”. The reviewer concerns that “*The topic is not new as well as the materials employed*”. In fact, we did not write clearly that the main task of this work is to tune defect engineering of oxides in the last version. Defect engineering has been considered as an effective strategy to tune the physical and chemical properties of materials and further broaden their applications stemmed from the versatile electronic properties of defective materials^{R1-R7}. As shown in this work, the defective oxides also exhibit high performance for photocatalytic hydrogen evolution, while they should have more potential applications. The main achievement of this work is that we realized the defect engineering of oxides at mild conditions by a simple approach.

Meanwhile, we fully agree that the oxide materials in our study (including TiO₂, ZnO, SnO₂ and CeO₂) have been intensely studied for many years as traditional material systems with a long history. Most of the recognitions on the oxides are confined to the stoichiometric oxides, while the nonstoichiometric oxides through defect engineering

exhibit superior performance in many fields^{R1,R2,R6-R9}. While the challenging issues is to control the defect effectively. In this work, the main task is to make the “old” oxides “sparkle” through the defect engineering.

In this article, we propose a lithium reduction strategy to controllably tune the defects in oxide materials at mild conditions with advantages of all-room-temperature processing, easiness to control, fastness, versatility and scalableness. Furthermore, the as-prepared defective TiO₂ nanopowders by lithium reduction strategy demonstrated superior photocatalytic performance. Indeed, we are not only reporting a new method to tune the defects in oxide materials, but also introducing new idea to solve the problems in defect engineering areas. Moreover, please note that the widely reported oxide materials in the past decades will not damage the novelty of this work; instead, the creation of lithium reduction strategy will greatly promote the final application of these oxide materials.

To summary, in this article, we propose a lithium reduction strategy to controllably tune the defects in oxide materials at mild conditions with many advantages. Till now, this defects engineering strategy has not yet been reported. Therefore, we firmly believe our results will attract lots of attention from science community.

Corresponding changes in page 2, line 6:

Our lithium reduction strategy to tune defects in oxide materials shows advantages like all-room-temperature processing, easiness to control, fastness, versatility and scalableness. As one of potential applications, the performance for photocatalytic hydrogen evolution of the defective TiO₂ was examined, with the hydrogen evolution rate being up to 41.8 mmol g⁻¹ h⁻¹ under one solar light irradiation, which is ~3 times higher than pristine nanoparticles. The strategy of tuning the defect oxides used in this work should be beneficial for the many related applications.

Corresponding changes in page 3, line 1:

The ability to tune the defect structure of oxide materials has been a major area of focus since their fundamental physical and chemical properties greatly rely on their defect structures¹⁻⁷.

Corresponding changes in page 3, line 17:

Meanwhile, the defect engineering is an effective way to tune the electronic structure of metal oxides, which is vital for many applications^{2,5,23-26}.

Corresponding changes in page 9, line 11:

As discussed in the introduction, the defective oxides usually exhibits the superior properties, thus they exhibits good performance in many applications^{2,5,23-26}. Thus we expect that the defective metal oxides fabricated in this work should have many applications. In the following, we examined the photocatalytic degradation of organic pollutants and hydrogen evolution of TiO₂ nanopowders as potential applications.

Reference:

- R1. Koketsu, T. *et al.* Reversible magnesium and aluminium ions insertion in cation-deficient anatase TiO₂. *Nat. Mater.* **16**, 1142-1148 (2017).
- R2. Tan, H. *et al.* Efficient and stable solution-processed planar perovskite solar cells via contact passivation. *Science* **355**, 722-726 (2017).
- R3. Kim, H. S. *et al.* Oxygen vacancies enhance pseudocapacitive charge storage properties of MoO_{3-x}. *Nat. Mater.* 4810 (2016).
- R4. Yang, W. S. *et al.* Iodide management in formamidinium-lead-halide-based perovskite layers for efficient solar cells. *Science* **356**, 1376 (2017).
- R5. Giordano, F. *et al.* Enhanced electronic properties in mesoporous TiO₂ via lithium doping for high-efficiency perovskite solar cells. *Nat. Commun.* **7**, 10379 (2016).
- R6. Gong, C. *et al.* Discovery of intrinsic ferromagnetism in two-dimensional van der Waals crystals. *Nature* **546**, 265-269 (2017).
- R7. Yang, Y. *et al.* Deciphering chemical order/disorder and material properties at the single-atom level. *Nature* **542**, 75-79 (2017).
- R8. Sambur, J. B. *et al.* Sub-particle reaction and photocurrent mapping to optimize catalyst-modified photoanodes. *Nature* **530**, 77-80 (2016).
- R9. Zhang, X. *et al.* Direction-specific van der Waals attraction between rutile TiO₂ nanocrystals. *Science* **356**, 434-437 (2017).

Comment 2: *The performance in the application could be an important aspect to evaluate the relevance of the work. The author should compare the photocatalytic hydrogen evolution rate that they achieved with the SoA.*

Response: We thank this reviewer for the excellent suggestion. We have added them in the revised manuscript.

Corresponding changes in page 12, line 8:

Furthermore, we also compare the photocatalytic hydrogen evolution activity of the lithium reduced TiO₂ nanoparticles with the previous reported works (Table S2),

indicating our defective TiO₂ by lithium reduction possesses superior photocatalytic activity.

Table S2. Comparison of photocatalytic hydrogen evolution activity with the available experiments

Catalyst	Hydrogen evolution rate (mmol g ⁻¹ h ⁻¹)	Light source	SoA	Refs
1% Au/TiO ₂	10.2	Iron halogenide mercury arc lamp, 250 W	6% CH ₃ OH	61
0.6% Pt/TiO ₂	10	AM 1.5 solar simulator	50% CH ₃ OH	18
1% PtO/TiO ₂	4.4	Xe lamp, 300 W	30% CH ₃ OH	62
1% Pt/TiO ₂	29	Xe lamp, 300 W	50% CH ₃ OH	63
1% Pt/TiO ₂	6.32	Xe arc lamp, 300 W	20% CH ₃ OH	64
1% Pt/TiO ₂	6.5	Xe lamp, 300 W	25% CH ₃ OH	65
1% Pt/TiO ₂	2.15	Xe arc lamp, 300 W	20% CH ₃ OH	66
0.5% Pt/TiO ₂	6.4	Hg lamp, 300 W	25% CH ₃ OH	20
1% Pt/TiO ₂	43.2	Xe lamp, 400 W	20% CH ₃ OH	19
1% Pt/TiO ₂	4	Xe lamp, 200 W	10% CH ₃ OH	67
0.5% Pt/TiO ₂	15	Xe lamp, 300 W	20% CH ₃ OH	31
1% Pt/TiO ₂	2	Xe lamp, 300 W	30% CH ₃ OH	68
0.57% Pd/TiO ₂	3.32	Xe lamp, 300 W	20% CH ₃ OH	8
1% Pt/TiO ₂	2.41	Xe lamp, 300 W	20% CH ₃ OH	69
0.5% Pt/TiO ₂	5.2	Xe lamp, 300 W	20% CH ₃ OH	70
1% Pt/TiO ₂	10.6	Xe lamp, 300 W	50% CH ₃ OH	71
0.5% Pt/TiO ₂	1.5	Xe lamp, 300 W	20% CH ₃ OH	72
1% Pt/TiO ₂	41.8	Xe lamp, 300 W	20% CH ₃ OH	This work

Reference:

61. Chiarello, G. L., Forni, L. & Selli, E. Photocatalytic hydrogen production by liquid- and gas-phase reforming of CH₃OH over flame-made TiO₂ and Au/TiO₂. *Catal. Today* **144**, 69-74 (2009).
62. Li, Y. H. *et al.* Unidirectional suppression of hydrogen oxidation on oxidized platinum clusters. *Nat. Commun.* **4**, 2500 (2013).
63. Cai, J. M. *et al.* In situ formation of disorder-engineered TiO₂(B)-anatase heterophase junction for enhanced photocatalytic hydrogen evolution. *ACS Appl. Mater. Interfaces* **7**, 24987-24992 (2015).
64. Tian, J., Leng, Y. H., Cui, H. Z. & Liu, H. Hydrogenated TiO₂ nanobelts as highly efficient photocatalytic organic dye degradation and hydrogen evolution photocatalyst. *J. Hazard. Mater.* **299**, 165-173 (2015).
65. Tan, H. Q. *et al.* A facile and versatile method for preparation of colored TiO₂ with enhanced solar-driven photocatalytic activity. *Nanoscale* **6**, 10216-10223 (2014).
66. Zheng, Z. K. *et al.* Hydrogenated titania: Synergy of surface modification and morphology improvement for enhanced photocatalytic activity. *Chem. Commun.* **48**, 5733-5735 (2012).
67. Li, L. *et al.* Sub-10 nm rutile titanium dioxide nanoparticles for efficient visible-light-driven photocatalytic hydrogen production. *Nat. Commun.* **6**, 5881 (2015).
68. Zhao, Z. *et al.* Effect of defects on photocatalytic activity of rutile TiO₂ nanorods. *Nano Res.* **12**, 4061-4071 (2015).
69. Hu, W. Y. *et al.* Facile strategy for controllable synthesis of stable mesoporous black TiO₂ hollow spheres with efficient solar-driven photocatalytic hydrogen evolution. *J. Mater. Chem. A* **4**, 7495-7502 (2016).
70. Zhu, G. L. *et al.* Black titania for superior photocatalytic hydrogen production and photoelectrochemical water splitting. *ChemCatChem* **7**, 2614-2619 (2015).
71. Wang, Y. T. *et al.* Hydrogenated cage-like titania hollow spherical photocatalysts for hydrogen evolution under simulated solar light irradiation. *ACS Appl. Mater. Inter.* **8**, 23006-23014 (2016).
72. Zhang, K. F. *et al.* Black N/H-TiO₂ nanoplates with a flower-like hierarchical architecture for photocatalytic hydrogen evolution. *ChemSusChem* **9**, 2841-2848 (2016).

Comment 3: *The authors show higher defect density and, for the same material, higher conductivity, they give some explanation in the conclusions of the paper, but the transport mechanism should be explained better, as the material is mesoporous.*

Response: We thank this reviewer for the insight. As shown in many previous works, the existence of the Ti^{3+} can enhance the conductivity^{R10-R13}. Here, the electronic states for the reduced TiO_2 with different O_v concentrations were explored in the new version. The oxygen vacancy will induce the excess electrons in the n-type metal oxides. As shown in Figure S8, when the system contains the oxygen vacancy, the excess electrons are mainly localized on the Ti atoms, which make Ti^{4+} into Ti^{3+} . Meanwhile the concentration of the Ti^{3+} is improved with the increase of the O_v . It is worth noting that the excess electrons can freely diffuse with a small energy barrier of 0.3 eV^{R14}, confirmed by the first-principles molecular dynamics^{R15,R16}. Thus the main reason for the high conductivity should come from the excess electrons (Ti^{3+}) induced by the O_v .

Corresponding changes in page 7, line 7:

It is well-known that the existence of the Ti^{3+} can enhance the conductivity of the TiO_2 ³³⁻³⁵. As shown in Figure S8, the existence of oxygen vacancies induces the excess electrons, which make the Ti^{4+} into Ti^{3+} . Meanwhile the concentration of Ti^{3+} greatly depends on the amount of the O_v . The excess electrons can freely hop at the room temperature³⁶⁻³⁸, which should be the main reason for the high conductivity of the reduced TiO_2 .

Figure S8. The side view of defect $\text{TiO}_2(101)$ with different subsurface O_v (blue circles) concentrations. Here, the 1O_v (a), 2O_v (b) and 4O_v (c) per formula were considered in our simulations, and the O_v stay in the subsurface. The spin density of the occupied states for different O_v concentrations is also shown in yellow and blue

contours for the two different phases. The red and light blue spheres represent O and Ti atoms, respectively. O_v s are shown in the blue circles.

Reference:

- R10. Tang, K. C. *et al.* Distinguishing oxygen vacancy electromigration and conductive filament formation in TiO_2 resistance switching using liquid electrolyte contacts. *Nano Lett.* **17**, 4390-4399 (2017).
- R11. Lu, X. J. *et al.* Conducting interface in oxide homojunction: Understanding of superior properties in black TiO_2 . *Nano Lett.* **16**, 5751-5755 (2016).
- R12. Amano, F., Nakata, M., Yamamoto, A. & Tanaka, T. Effect of Ti^{3+} ions and conduction band electrons on photocatalytic and photoelectrochemical activity of rutile titania for water oxidation. *J. Phys. Chem. C* **120**, 6467-6474 (2016).
- R13. Deskins, N. A. & Dupuis, M. Electron transport via polaron hopping in bulk TiO_2 : A density functional theory characterization. *Phys. Rev. B* **75**, 19521219 (2007).
- R14. Kowalski, P. M., Camellone, M. F., Nair, N. N., Meyer, B. & Marx, D. Charge localization dynamics induced by oxygen vacancies on the titania $TiO_2(110)$ surface. *Phys. Rev. Lett.* **105**, 146405 (2010).
- R15. Spreafico, C. & VandeVondele, J. The nature of excess electrons in anatase and rutile from hybrid DFT and RPA. *Phys. Chem. Chem. Phys.* **16**, 26144-26152 (2014).

Reviewer #2:

Comment 1: *Authors investigate a room temperature reduction process for binary oxides using Li metal and the property, especially photocatalytic property of TiO_{2-x} . The results show oxide powders (TiO_2 , ZnO , SnO_2 and CeO_2) were reduced by room temperature grinding with Li metal powders in dimethyl carbonate. The obtained TiO_{2-x} exhibited high photocatalytic properties. This method seems a new facile way to reduce oxides.*

Response: We thank this reviewer and feel grateful that the reviewer has carefully read our manuscript.

Comment 2: *However, there is not so much sufficient impact. The metal reduction using Al at around 500 degC has been reported (high temperature but safe "ref: Energy Environ. Sci. 2013, 6, 3007"), and laser reduction has also been reported (not good for bulk materials but room temperature ambient atmosphere "ref: J. Mater. Chem. A 2014, 2, 6762").*

Response: We thank the reviewer for pointing out these important references from Prof. Huang's group and Prof. Nakajima's group. We apologize for not making the novelty of this study clear enough in our manuscript and would like to give more discussion on our scientific contribution. We strongly agree with this reviewer that defects engineering have become a very hot and important topic in science community, and some recent breakthroughs on the defects engineering of TiO_2 materials have been recently reported, for example, Nakajima *et al.* reported defective TiO_2 material by high energy laser irradiation, which provides us very important strategy to tune the defects in oxide materials^{R17,R18}. In addition, chemical reduction route to generate defects in TiO_2 has also been reported as an effective strategy, like high temperature reduction by H_2 , Mg, Al and other reductive chemicals, however, these reduction often required high-temperature, pressure control and/or long-term processing (Table R1), which usually lead to crystal structural and morphological changes of nanomaterials. Furthermore, it is difficult to control the concentration of defects by such high temperature reduction. We compared our reduction method with previous reported methods in Table R1 as attached below.

To summary, we not only reported defective TiO_2 photocatalyst with high activity and stability, but also developed new strategy for controllable preparation of defective oxides with remarkable advantages that have not been reported before in other works. The Li reduction strategy in this report significantly showed following advantages:

1. All-room-temperature processing: Most of the oxides can be reduced by Li metal at room-temperature due to its high reductive activity; meanwhile, we can engineer the defects in oxides without changing their morphology and crystal structure at room-temperature.
2. Easiness to control: It is one of the most promising but challenging question to easily tune the content of defects in oxides in the field of defects engineering. Any breakthrough could contribute not only to the photocatalysts, but to the whole field of defect engineering. Although great effect has been made, as far as we know, it is still a big challenge for this field. In our lithium reduction strategy, we have successfully tuned the content of defects in oxides by simply adjusting the weight ratio of lithium powder to oxides, showing significant advantages than the previous reports (Figure R1).
3. Fastness, versatility and scalableness: As demonstrated in our manuscript, a series of defective oxides have been prepared by simply mixing and grinding the target oxide with lithium powder in a glove box in a few hours without expensive equipment. Meanwhile, the amount of the defective oxides can be easily scaled up as the commercialized lithium metal powder has been supplied in large quantities.

Therefore, we believe this work opens up a new methodology different from previous ones for defects engineering of oxide materials, which will have profound influence on future defective oxides research.

We added more supporting information:

Table R1. Technical comparison of lithium reduction strategy with the previous reported methods

Method	Conditions	Equipment	Catalyst	Refs.
Aluminium reduction	0.5 Pa, Al@800 °C and TiO ₂ @500 °C, 6 h	A two-zone tube furnace	TiO ₂ nanopowders	R16
Pulsed ultraviolet laser irradiation	500 Pa, Laser irradiation, 150 s	Pulsed KrF laser	TiO ₂ thin film	R17
Lithium reduction	Room-temperature, Atmospheric Pressure, 1 h	Glove box	TiO ₂ nanopowders	This work

Corresponding changes in page 3, line 12:

Reductive metals have also been introduced to tune the defects in TiO₂ material at high temperatures, such Mg and Al^{19,20}. In addition, Nakajima *et al.* reported defective TiO₂ material by high energy laser irradiation, which provides us another strategy to tune the defects in oxide materials^{21,22}. Nevertheless, it keeps challenging to controllably engineer the defects in oxide materials at ambient conditions. Meanwhile, the defect engineering is an effective way to tune the electronic structure of metal oxides, which is vital for many applications^{2,5,23-26}.

Corresponding changes in page 15, line 5:

We developed a lithium reduction approach to conveniently implant defects in oxide materials with advantages of all-room-temperature processing, easiness to control, fastness, versatility and scalability. Besides, the defects in TiO₂ nanoparticles were successfully tuned, which demonstrated significantly enhanced photocatalytic activity and stability in degradation and hydrogen evolution reactions after lithium reduction treatment. In addition, a series of oxide materials, including ZnO, SnO₂ and CeO₂, have also been successfully modified through implanted defects by a similar Li reduction process. Moreover, it is firmly believed that the properties of other functional oxide materials can be optimized by the proposed lithium reduction approach, which can be applied to many areas such as environmental protection, microelectronics, energy conversion and storage.

Figure R1. Photograph of oxide nanopowders reduced by different weight ratio of lithium to oxides. a, TiO_2 . b, ZnO. c, SnO_2 . d, CeO_2 .

Reference:

- R16. Wang, Z. *et al.* Visible-light photocatalytic, solar thermal and photoelectrochemical properties of aluminium-reduced black titania. *Energy Environ. Sci.* **6**, 3007-3014 (2013).
- R17. Nakajima, T., Nakamura, T., Shinoda, K. & Tsuchiya, T. Rapid formation of black titania photoanodes: Pulsed laser-induced oxygen release and enhanced solar water splitting efficiency. *J. Mater. Chem. A* **2**, 6762-6771 (2014).
- R18. Nakajima, T., Tsuchiya, T. & Kumagai, T. Pulsed laser-induced oxygen deficiency at TiO_2 surface: Anomalous structure and electrical transport properties. *J. Solid State Chem.* **182**, 2560-2565 (2009).

Comment 3: *The experimental procedure of Li metal method is very simple, however, it needs strictly managed environment to safely treat Li metal powders. Therefore, this method also has merit and demerit as well as the other methods.*

Response: We thank this reviewer for the valuable comments. The lithium reduction of materials has been widely applied not only for scientific study but also in industry

production. For example, in industrial, Li metal is widely used as a flux for welding or soldering metals to promote the fusing of metal during the process, eliminates the forming of oxides by absorbing impurities^{R19}. In addition, Li metal also has important applications for energy storage^{R20}.

We agree that Li metal should be strictly managed due to its high activity, while it only requires a glove box in our experiment. Firstly, the lithium powders used in this work were massively produced in industry and treated by surface passivation, showing a high degree of security. Besides, our lithium powers were put into a sealed glove box full of argon for the safe storage and usage. Lastly, just a small amount of lithium powders (~3 wt%) were required in our method, meanwhile, the reaction product is a non-hazardous lithium oxides which can be easily removed by washing with water. In order to clearly show the safety and advance of our lithium reduction strategy, the detailed process for the preparation of defective TiO₂ nanopowders reduced by 5 wt% Li was recorded and revealed in Figure R2.

We gave more discussion on this point. Corresponding changes in page 26, line 13:

It is noted that although highly active Li metal powders were used in the experiment, there are no danger in the whole experimental process in regular operation. Firstly, the lithium powders used in this work were massively produced in industry and treated by surface passivation, showing a high degree of security. Besides, the lithium powers were put into a sealed glove box full of argon for the safe storage and usage. Lastly, just a small amount of lithium powders (less than 5 wt%) were required by our method.

Figure R2. Photograph of detailed process for the preparation of defective TiO₂ nanopowders reduced by 5 wt% Li.

Reference:

- R19. Garrett, D. E. Handbook of lithium and natural calcium chloride. *Academic Press*, 2004, p200.
- R20. Wadia, C., Albertus, P., & Srinivasan, V. Resource constraints on the battery energy storage potential for grid and transportation applications. *J. Power Sources*. **196**, 1593-1598 (2011).

Comment 4: *I feel a critical problem about the manuscript itself. Most important point is mechanism of reduction using Li at room temperature just by grinding in dimethyl carbonate. However, it is not explained at all in the text.*

Response: We thank this viewer for his/her critical comments. We would like to apologize for not making our mechanism clear enough in our manuscript, which might cause some confusion on the understanding of our work. The detailed

mechanism has been added in our revised manuscript. Firstly, the ΔG for reaction $4\text{Li}+\text{TiO}_2\rightarrow\text{Ti}+2\text{Li}_2\text{O}$ is $-233.6\text{ kJ mol}^{-1}$ (or 2.43 eV/atom) at 298 K , indicating the reaction can occur spontaneously at room-temperature. Thus the lithium metal can draw out O from TiO_2 to generate Li_2O as well as defective TiO_2 in the grinding process. Secondly, the lithium was continuously extended during the grinding process because of the high ductility, resulting in the constant exposure of fresh lithium to react with the oxides, so that the lithium metal would react fully. Lastly, the purpose of adding dimethyl carbonate is to allow the lithium metal to be in fully contact with the oxides during the experiment. For a detailed explanation, please see:

Corresponding changes in page 4, line 9:

For example, the ΔG for reaction $4\text{Li}+\text{TiO}_2\rightarrow\text{Ti}+2\text{Li}_2\text{O}$ is $-233.6\text{ kJ mol}^{-1}$ at 298 K , indicating the reaction can occur spontaneously at room-temperature.

Corresponding changes in page 8, line 14:

From the above results, we can see that uniform defects have been successfully implanted at the surface of the oxide nanopowders by our lithium reduction strategy, which can be ascribed to the following three reasons. Firstly, the lithium metal can act with the oxides at room-temperature according to the Ellingham diagrams and our calculated results. Thus the lithium metal can draw out O from the oxides to generate Li_2O as well as defective oxides in the grinding process. Secondly, the lithium can be continuously extended during the grinding process because of its high ductility, resulting in the constant exposure of fresh lithium to react with the oxides, so that the lithium metal would react fully. Lastly, the lithium powders and oxide nanopowders can be uniformly mixed with the adding of dimethyl carbonate, which is beneficial for the uniform implantation of defects at the surface of the oxide nanopowders in the grinding process.

Comment 5: *The analysis and characterization of the property for the obtained powder are no problem, and these information are not important (e.g. it has been already reported in so many papers that reduced TiO_{2-x} has several times higher photocatalytic property than pristine TiO_2). But there is hardly sufficient value, if authors make the mechanism clear.*

Response: We thank the reviewer for this suggestion. We have added the detailed mechanism in the revised manuscript.

Corresponding changes in page 14, line 11:

From the above results, it can be clearly seen that the lithium reduced TiO₂ nanoparticles show significantly enhanced photocatalytic property compared with the pristine TiO₂, which may be correlated with the greatly increased light absorption, improved conductivity, surface disorder layer, implanted oxygen vacancies and generated Ti³⁺. In particular, the improved light absorption and photocatalytic properties are attributed to the Ti³⁺ of the defective TiO₂. Furthermore, metallic conduction can be achieved at the crystalline-amorphous interface of the defective TiO₂ nanoparticles, which would enhance the electron transport property of TiO₂³⁴. Another important factor is the implanted oxygen vacancies and/or surface disorder, which also plays a great role in increasing the photocatalytic activity. The donor density of TiO₂ is enhanced by introducing oxygen vacancies as they can act as electron donors⁴⁹, which would improve charge transport and shift the Fermi level toward conduction band⁵⁰, facilitating charge separation and improving the IPCE in UV region⁵¹. Finally, the generated Ti³⁺ in the defective TiO₂ can reduce the recombination of the photogenerated electron-hole pairs and thus improve the photocatalytic activity of the lithium reduced TiO₂ nanoparticles¹⁹.

Comment 6: *Why is it possible to reduce oxides just by grinding with very small amount Li metal powders?*

Response: We thank the reviewer for the comment. We have checked many times in our experiments that only small amount of Li is required to obtain the defects. This can be explained by considering the thermal dynamics and kinetics of the lithium reduction reactions.

First of all, the reduction has enough driving force at room temperature. The ΔG for reaction $4\text{Li} + \text{TiO}_2 \rightarrow \text{Ti} + 2\text{Li}_2\text{O}$ is -233.6 KJ/mol (or 2.43 eV/atom), indicating the reaction can occur spontaneously at room-temperature. Furthermore, the calculated reaction enthalpy for $x\text{Li} + \text{TiO}_2 \rightarrow \text{TiO}_{2-x/2} + x/2 \text{Li}_2\text{O}$ is -4.03 eV (Table S1), where the negative value means an exothermic process. Thus lithium metal can capture O from TiO₂ to generate Li₂O as well as defective TiO₂ in the grinding process. In addition, the relative atomic mass of lithium is relatively small (6.941). According to reaction equation $x\text{Li} + \text{TiO}_2 \rightarrow \text{TiO}_{2-x/2} + x/2 \text{Li}_2\text{O}$, when 5 wt% Li was added to TiO₂, the theoretical product was TiO_{1.712} after completion of the reaction. In other words, the content of oxygen defect in the final product was 14.4%. So a small amount of lithium metal powder can be effective in reducing the oxides to produce defective oxides. Lastly, the lithium atoms have high diffusion speed in oxide materials; also benefit the controlled formation of oxygen defects.

We give more discussion on this point. Corresponding changes in page 8, line 14:

From the above results, we can see that uniform defects have been successfully implanted at the surface of the oxide nanopowders by our lithium reduction strategy, which can be ascribed to several reasons. Firstly, the lithium metal can act with the oxides at room-temperature according to the Ellingham diagrams and our calculated results. Thus the lithium metal can draw out O from the oxides to generate Li_2O as well as defective oxides in the grinding process. Secondly, the lithium can be continuously extended during the grinding process because of its high ductility, resulting in the constant exposure of fresh lithium to react with the oxides, so that the lithium metal would react fully. Lastly, the lithium powders and oxide nanopowders can be uniformly mixed with the adding of dimethyl carbonate, which is beneficial for the uniform implantation of defects at the surface of the oxide nanopowders in the grinding process.

Comment 7: *Why does Li capture oxygen preferentially from oxides, not from organic solvent?*

Response: We thank the reviewer for the comment. In fact, the solvents have high chemical stability with Lithium metal and have been applied widely as components of electrolytes in Li-ion batteries. First of all, based on the above thermodynamic calculations, lithium metal can capture O from TiO_2 to generate Li_2O and defective TiO_2 at room-temperature. Our first-principles calculations show that the formation energy of the oxygen vacancy in TiO_2 is 3.77 eV per oxygen, while the corresponding formation energy of losing one oxygen in DMC is 4.80 eV per oxygen, which is 1.03 eV larger than the corresponding one through capturing the oxygen from TiO_2 . Such results clearly suggest the reduction of DMC is more difficult to occur compared with TiO_2 , and the Li prefers to capture O from the TiO_2 instead of organic solvent. At last, DMC as a dispersant is widely used in lithium-ion battery areas as an important component part of the organic electrolyte^{R21,R22}, which means this solvent has good stability with lithium metal at room-temperature.

Figure R3. The model used for reduction of dimethyl carbonate (DMC). Left is the

perfect DMC, and the right one is the reduced DMC. The brown, red and white spheres stand for C, O and H atoms, respectively.

Corresponding changes in page 8, line 9:

Our first-principles calculations show that the formation energy of the oxygen vacancy in TiO_2 is 3.77 eV per oxygen, while the corresponding formation energy of losing one oxygen atom in DMC is 4.80 eV per oxygen, which is 1.03 eV larger than the corresponding one through capturing the oxygen from TiO_2 . Thus the Li prefers to capture oxygen from the TiO_2 instead of the organic molecules.

Reference:

- R21. Rodrigues, M. T. F. et al. A materials perspective on Li-ion batteries at extreme temperatures. *Nature Energy* **2**, 17108 (2017).
- R22. Song, H. et al. Anomalous decrease in structural disorder due to charge redistribution in Cr-doped $\text{Li}_4\text{Ti}_5\text{O}_{12}$ negative-electrode materials for high-rate Li-ion batteries. *Energy Environ. Sci.* **5**, 9903-9913 (2012).

Comment 8: *Why is such small amount of Li powder enough for homogeneous reduction of oxide powders?*

Response: We thank the reviewer for the comment. Firstly, the lithium metal was continuously extended during the grinding process because of its high ductility. This led to the constant exposure of fresh lithium to react with the oxide, so that lithium would react fully and generate defective oxide. Secondly, our oxide materials are approximately spherical and they can scroll in the grinding process, so the surface of the nanopowders has an equal opportunity to react with the lithium to form an oxide material with evenly surface defects distribution. Furthermore, as shown in Figure 2e, defects are mainly distributed at the surface regions of the defective oxide nanopowders. Thirdly, although the oxygen vacancy is produced in the surface region, it is well-known that O_v in anatase prefers to diffuse from the surface to the subsurface and deep layers^{R23-R25}. As shown in the Figure S22, our first-principles calculations suggested that the O_v initially created on the surface can easily diffuse from the surface to the subsurface, as reported in previous experiment and theory^{R23-R25}. Then, the surface O_v could diffuse from the surface into the subsurface with a small diffusion barrier (~ 0.7 eV), and the relatively small energy barrier could help the homogeneous distribution of O_v .

Corresponding changes in page 9, line 3:

In addition, the reaction between the Li and TiO₂ occur at the interface region, which captures the oxygen atoms of the TiO₂ surface. Although the oxygen vacancies are initially generated at the surface regions, they prefer to diffuse into the subsurface or the inner layer (Figure S22). The subsurface O_v is more stable by about 0.13 eV than the one in surface, as suggested in the previous experiment and theory³⁹⁻⁴¹. The calculated energy barrier is about 0.7 eV for the diffusion barrier (Figure S22), which indicates that the oxygen vacancy has the tendency to diffuse from the surface to the inner layer region at the room temperature.

Figure S22. The reaction pathway of the O_v diffusion from surface (the left one) to subsurface (the right one). The total energy of surface O_v is set as the zero energy for reference. The light blue and red balls are Ti atoms and O atoms, respectively. The yellow ball stands for the moving oxygen atom during the O_v diffusion. The blue circle denotes the O_v defect. The transition states are shown on the up panel and the stable O_v sites are shown on the down panel.

Reference:

- R23. Cheng, H. Z. & Selloni, A. Energetics and diffusion of intrinsic surface and subsurface defects on anatase TiO₂(101). *J. Chem. Phys.* **131**, 54703 (2009).

R24. He, Y. B., Dulub, O., Cheng, H. Z., Selloni, A. & Diebold, U. Evidence for the predominance of subsurface defects on reduced anatase $\text{TiO}_2(101)$. *Phys. Rev. Lett.* **102**, 10610510 (2009).

R25. Cheng, H. Z. & Selloni, A. Surface and subsurface oxygen vacancies in anatase TiO_2 and differences with rutile. *Phys. Rev. B* **79**, 0921019 (2009).

Comment 9: *How about the effect of Li metal powder size on the reduction?*

Response: We thank the reviewer for the very helpful suggestion. We believe that the size of lithium powder has little effect on the reduction of oxides. Commercial lithium powder was used for our experiment with diameter of about 30 μm (Figure S11). Unfortunately, we have been looking for a long time but still haven't bought other sizes of lithium powder. In order to investigate the size effect of lithium on the reduction, we selected lithium powder and lithium piece as raw materials. Figure S12 shows the TiO_2 powder after grinding with 5 wt% Li powder (left) and Li piece (right). It is obvious that the defective TiO_2 produced with Li powder or Li piece exhibited the same color depth, indicating their defective content is consistent. In summary, we believe that the size of lithium powder has no effect on the reduction of oxides. In fact, the most important factor for this reaction is the content of lithium.

Corresponding changes in page 8, line 1:

In order to investigate the effect of Li metal powder size on the reduction of oxides, we prepared defective TiO_2 nanopowders deduced by 5 wt% Li powders and Li piece. Here, the diameter of the Li powders we used in the experiment is about 30 μm (Figure S11). As shown in Figure S12, it is apparent that the defective TiO_2 produced with Li powder or Li piece exhibited the same color depth, indicating their defective content is consistent.

Figure S11. SEM micrograph of lithium powders.

Figure S12. Photograph of TiO₂ nanopowders reduced by 5 wt% Li powders and Li piece respectively.

Comment 10: *How does the amount of solvent influence the reduction?*

Response: We thank the reviewer for the suggestion. Again, we conformed that the solvents have not join the reaction and mainly function to disperse the particles. In response to this comments, we performed new experiments to tune the amount of solvent added in the experiment. As shown in Figure S13, the weight ratio of oxides to DMC is 1:10, 1:20 and 1:30, respectively. It can be seen that no significant changes can be observed on the color of the products as the solvent content changes.

Corresponding changes in page 8, line 6:

In addition, we also compared different content of dispersant (DMC) on the reduction of TiO₂ nanopowders. As shown in Figure S13, it can be seen that the color of the defective TiO₂ did not change with the dispersant content change from 1:10, 1:20 to 1:30.

Figure S13. Photograph of TiO₂ nanopowders reduced by 5 wt% Li powders with different weight ratio of oxides to DMC.

Comment 11: *If the reduction of oxide proceed only at the contact interface with the Li powders, these points must be clear. If one Li particle has contact with an oxide particle and then the Li is simultaneously fully oxidized, it would be difficult to make homogeneously reduced oxide powders.*

Response: We thank the reviewer for the very helpful comment. First of all, as the response in the Comment 8 rose by the same referee, although oxygen vacancies are initially formed on the anatase TiO₂ surface, they are energetically stabilized on the subsurface and the inner layer^{R24}. This is the main reason why the oxygen vacancy on anatase surface is seldom observed while widely exist in rutile^{R25}. Our first-principles calculations show the diffusion barrier of oxygen vacancy from the surface to the subsurface is about 0.7 eV (see Figure S22), which agrees well with the previous works^{R23,R25}. Therefore, the initially created O_v on the anatase surface will easily diffuse into the subsurface because of the lower barrier for the diffusion. Consequently, the O_v will concentrate on the subsurface homogeneously.

There are several other reasons accounting for it. First, lithium was a reducing agent taking O from the oxide to produce a defective oxide material in our method.

Especially, DMC was added as a dispersant in the experiment to allow sufficient contact between the lithium and the oxide to ensure uniformity of the reaction. Second, theoretically, lithium reacts with the oxide generating lithium oxide, which may hinder further reduction; however, in practice, the lithium was continuously extended during the grinding process because of the high ductility. This led to the constant exposure of fresh lithium to react with the oxide, so that lithium would react fully. Third, our oxide material is approximately spherical, so the surface of the nanopowders has an equal opportunity to react with the lithium to form an oxide material with evenly surface defects distribution.

Corresponding changes in page 9, line 3:

In addition, the reaction between the Li and TiO_2 occur at the interface region, which captures the oxygen atoms of the TiO_2 surface. Although the oxygen vacancies are initially generated at the surface regions, they prefer to diffuse into the subsurface or the inner layer (Figure S22). The subsurface O_v is more stable by about 0.13 eV than the one in surface, as suggested in the previous experiment and theory³⁹⁻⁴¹. The calculated energy barrier is about 0.7 eV for the diffusion barrier (Figure S22), which indicates that the oxygen vacancy has the tendency to diffuse from the surface to the inner layer region at the room temperature.

Figure S22. The reaction pathway of the O_v diffusion from surface (the left one) to

subsurface (the right one). The total energy of surface O_v is set as the zero energy for reference. The light blue and red balls are Ti atoms and O atoms, respectively. The yellow ball stands for the moving oxygen atom during the O_v diffusion. The blue circle denotes the O_v defect. The transition states are shown on the up panel and the stable O_v sites are shown on the down panel.

Reference:

- R23. Cheng, H. Z. & Selloni, A. Energetics and diffusion of intrinsic surface and subsurface defects on anatase $TiO_2(101)$. *J. Chem. Phys.* **131**, 54703 (2009).
- R24. He, Y. B., Dulub, O., Cheng, H. Z., Selloni, A. & Diebold, U. Evidence for the predominance of subsurface defects on reduced anatase $TiO_2(101)$. *Phys. Rev. Lett.* **102**, 10610510 (2009).
- R25. Cheng, H. Z. & Selloni, A. Surface and subsurface oxygen vacancies in anatase TiO_2 and differences with rutile. *Phys. Rev. B* **79**, 0921019 (2009).

Second round of reviews:

Reviewers' comments:

Reviewer #1 (Remarks to the Author):

The authors have clarified the points of concern from my point of view, I therefore recommend for publication.

Reviewer #2 (Remarks to the Author):

The authors modified the manuscript corresponding to my concerns one by one, and the manuscript was much improved. But I request more evaluations for the obtained reduced particles before the publication.

1. Only the local structural information can be understood from the TEM data, indicating the defective structure. How about the averaged structure? It can be observed from some diffuse scattering of XRD or some other analyses? Readers do want to know this powder is not the mixture of TiO_2 and TiO_{2-x} but the homogeneous TiO_{2-x} . Even from the XRD data, it can be discusses.

2. How do you carry out the grinding? By hand? The details must be written in the Methode section. This is important information in this manuscript.

3. The contact reaction between the Li and TiO_2 powders during the grinding (by hand?) means that this reaction proceed quite fast time scale. And then the oxygen diffusion in the TiO_{2-x} just after the capturing oxygen by the Li is also very smooth. Are these really reasonable? I'm not sure how many times one TiO_2 particle has chences to contact with the fresh Li powders. This point is very critical point in this reaction process.

Response Letter

Reviewer #1:

Comment 1: *The authors have clarified the points of concern from my point of view, I therefore recommend for publication.*

Response: We thank this reviewer for suggesting acceptance of our work.

Reviewer #2:

Comment 1: *Only the local structural information can be understood from the TEM data, indicating the defective structure. How about the averaged structure? It can be observed from some diffuse scattering of XRD or some other analyses? Readers do want to know this powder is not the mixture of TiO_2 and TiO_{2-x} but the homogeneous TiO_{2-x} . Even from the XRD data, it can be discusses.*

Response: We thank this reviewer for the comments. We agree with this reviewer that the TEM images can only exhibit local structures of single TiO_2 nanopowder, this is the nature of many materials characterization methods like TEM, SEM and AFM. However, such characterizations still provide important information since the particles observed in TEM were chosen randomly and we confirmed the structure on multiple samples. For XRD analysis, it is very difficult to determine whether lithium reduced TiO_2 nanopowders is *homogeneous TiO_{2-x}* or *mixture of TiO_2 and TiO_{2-x}* , since TiO_{2-x} material doesn't provide new XRD peaks, and the results are limited by the accuracy of XRD analysis. We can speculate that the lithium reduced TiO_2 nanopowders possess defective structure according to the following reasons: (1) Besides TEM, the defect structures have also been confirmed by XPS spectra and EPR spectra (Figure 2 and S10), which provided the average analysis of the materials; (2) The defective oxide powders have demonstrated significantly enhanced photocatalytic property after lithium reduction. In addition, the materials experienced uniform color changes from white to black; (3) Our experimental procedure ensures that all substrates are evenly involved in the reaction. First, we chose oxide nanoparticles and Li powders instead of micro oxide powders and Li foil as the substrates to avoid the possible uneven contact. Second, we added solvent (DMC) to enhance their contact while grinding. Last, we applied long time (~1 h) grinding process to allow the oxide particles evenly attach and react with the lithium powders. (4) Our experimental and theoretical discussion also pointed out that the Li atoms have a high reaction rate with oxides; that is to say, the defect implanting reaction is fast in dynamic, which further assistant the formation of uniform defects in the nanoparticles.

Considering the significant improved material performance, we believed that on average, there are rich amount of active defects been implanted in the materials, and our future work will focus on developing more advanced grinding methods like machine-grinding or ball-milling in order to make more uniform materials. To address

this comment, we add more discussion in the revised manuscript.

Corresponding changes in page 15, line 22:

Thus, we can speculate that almost all of our lithium reduced oxide nanopowders have defective structure based on the following reasons: (1) Besides TEM, the defect structures have also been confirmed by XPS spectra and EPR spectra (Figure 2 and S10), which provided the average analysis of the materials; (2) The defective oxide powders have demonstrated significantly enhanced photocatalytic property after lithium reduction. In addition, the materials experienced uniform color changes from white to black; (3) Our experimental procedure ensures that all substrates are evenly involved in the reaction. First, we chose oxide nanoparticles and Li powders instead of micro oxide powders and Li foil as the substrates to avoid the possible uneven contact. Second, we added solvent (DMC) to enhance their contact while grinding. Last, we applied long time (~1 h) grinding process to allow the oxide particles evenly attach and react with the lithium powders. (4) Our experimental and theoretical discussion also pointed out that the Li atoms have a high reaction rate with oxides; that is to say, the defect implanting reaction is fast in dynamic, which further assistant the formation of uniform defects in the nanoparticles.

Comment 2: *How do you carry out the grinding? By hand? The details must be written in the Method section. This is important information in this manuscript.*

Response: We thank the reviewer for this comment. We did the powder grinding by hand in our experimental process, which has been added in the revised manuscript. We give more details and discussions to address this comment.

Corresponding changes in page 28, line 8:

The materials were carefully mixed by hand grinding with speed of about 2 laps per second for 1 h and took out to dissolve the generated lithium oxide by dilute HCl.

Comment 3: *The contact reaction between the Li and TiO₂ powders during the grinding (by hand?) means that this reaction proceeds quite fast time scale. And then the oxygen diffusion in the TiO_{2-x} just after the capturing oxygen by the Li is also very smooth. Are these really reasonable? I'm not sure how many times one TiO₂ particle has chances to contact with the fresh Li powders. This point is very critical point in this*

reaction process.

Response: We thank the reviewer for this comment. This valuable comment inspired us to a deeper thinking of the reaction mechanisms. We agree that the contact reaction between the Li and TiO₂ powders proceeds very fast in principle, since Li can diffuse fast in the oxides. However, in our practical experiments, the process becomes much slower since lithium oxide passivation layers (like Li₂O) formed at the interface between Li and defective oxide powders, buffering the reaction between Li and oxides. Therefore, only a small amount of Li diffused into oxides while Li and oxide particle surface contacted. However, with the hand grinding continuous, the oxide passivation layers was destroyed by the shearing and friction forces between particles, with fresh lithium exposed to react with fresh oxides surfaces. In long-time grinding process, the lithium oxide passivation layer was cyclically formed and destroyed, and the lithium finally fully reacted with oxide powders, resulting in the formation of uniform defective oxide powders. We observed from Figure S1 that the color of the mixed powders turned darker after long time grinding, indicating that the defects formed with long time reactions. In our future study, ball milling or machine grinding can be introduced to speed up the reactions, making the process more controllable and efficient. In order to make this point clear, we have added more discussion in the revised manuscript.

Corresponding changes in page 15, line 7:

We note that the contact reaction between the Li and TiO₂ powders proceeds very fast in principle, since Li can diffuse fast in the oxides. However, we observed from Figure S1 that the color of the mixed powders turned darker after long time grinding, indicating that the defects formed with long time reactions. This is because lithium oxide passivation layers (like Li₂O) formed at the interface between Li and defective oxide powders, buffering the reaction between Li and oxides. Therefore, only a small amount of Li diffused into oxides while Li and oxide particle surface contacted. However, with the hand grinding continuous, the oxide passivation layers was destroyed by the shearing and friction forces between particles, with fresh lithium exposed to react with fresh oxides surfaces. In long-time grinding process, the lithium oxide passivation layer was cyclically formed and destroyed, and the lithium finally fully reacted with oxide powders, resulting in the formation of uniform defective oxide powders. While we herein use hand grinding to demonstrate the successful Li-reduction and defect-implanting chemistry, ball milling or machine grinding can be

introduced to speed up the reactions, making the process more controllable and efficient in the future study.

Corresponding changes in page 33, line 2:

Figure S1. Photograph of detailed process for the preparation of defective TiO₂ nanopowders reduced by 5 wt% Li.